# Synergism between two BLA-to-BNST pathways for appropriate expression of anxiety-like behaviors in male mice

Ren-Wen Han [1,2] ✉, Zi-Yi Zhang[1,3], Chen Jiao[1,3], Ze-Yu Hu[1,3] & Bing-Xing Pan [1,2] ✉

Understanding how distinct functional circuits are coordinated to fine-tune mood and behavior is of fundamental importance. Here, we observe that within the dense projections from basolateral amygdala (BLA) to bed nucleus of stria terminalis (BNST), there are two functionally opposing pathways orchestrated to enable contextually appropriate expression of anxiety-like behaviors in male mice. Specifically, the anterior BLA neurons predominantly innervate the anterodorsal BNST (adBNST), while their posterior counterparts send massive fibers to oval BNST (ovBNST) with moderate to adBNST. Opto-genetic activation of the anterior and posterior BLA inputs oppositely regulated the activity of adBNST neurons and anxiety-like behaviors, via disengaging and engaging the inhibitory ovBNST-to-adBNST microcircuit, respectively. Importantly, the two pathways exhibited synchronized but opposite responses to both anxiolytic and anxiogenic stimuli, partially due to their mutual inhibition within BLA and the different inputs they receive. These findings reveal synergistic interactions between two BLA-to-BNST pathways for appropriate anxiety expression with ongoing environmental demands.

Adapting to a dynamic environment necessitates the capacity to generate suitable emotional and behavioral responses, which relies on the coordination of diverse brain regions that are anatomically interconnected and functionally relevant[1–7]. Anxiety, as a common negative mood, emerges in response to possible threats and uncertainties[8]. It is regulated by an array of functionally distinct neuronal circuits encompassing limbic and extra-limbic regions[4,5,8–10]. Nonetheless, how these circuits are coordinated to enable contextually appropriate expression of anxiety remains largely unexplored.

The amygdala and BNST are two anatomically adjacent nuclei in the limbic system that are intricately interconnected and crucial in regulating moods such as anxiety[4,5,9,11–16]. Within each nucleus, there are more than ten subnuclei that possess different neuronal compositions and play distinct functional roles in mood regulation[13–19]. Furthermore, the specific neuronal subpopulations within these subnuclei

also have varying roles depending on their connections with upstream and downstream regions[9,11,12,15–18,20,21]. For instance, while optogenetic activation of BLA projections to lateral central amygdala (CeL) decreases the anxiety-like behavior in mice[22], activating those to medial prefrontal cortex (mPFC) promotes anxiety[23]. Despite the recognized significance of the interplay between BLA and BNST in anxiety regulation[17,24–26], it remains unclear about how the functionally distinct neurons in the BLA and BNST subnuclei collaborate to enable appropriate anxiety expression in response to environmental demands.

To address this issue, we systematically analyzed the connections made by projection neurons (PNs) in anterior and posterior BLA subdivisions (aBLA and pBLA which have opposing roles in emotion regulation)[27–33] onto the oval and anterodorsal BNST (ovBNST/ adBNST, two dorsal BNST subnuclei shown to oppositely regulate

[1]Laboratory of Fear and Anxiety Disorders, Institute of Biomedical Innovation, Jiangxi Medical College, Nanchang University, Nanchang 330031, China.
[2]School of Basic Medical Sciences, Jiangxi Medical College, Nanchang University, Nanchang 330031, China. [3]College of Life Science, Nanchang University, Nanchang 330031, China. ✉e-mail: hanrw@ncu.edu.cn; panbingxing@ncu.edu.cn

anxiety)[25,34–36]. We also explored the role of these connections in anxiety regulation and examined their potential interplay in processing anxiety-relevant information in the dynamic environment. Our findings revealed that the aBLA and pBLA PNs exhibit distinct patterns of projection to the adBNST and ovBNST, forming two functionally opposing pathways for anxiety regulation. Notably, these two parallel pathways not only receive different upstream afferents but interact extensively at both the start BLA and target dBNST regions by recruiting local inhibitory networks. The interactions between the two BLA-to-dBNST pathways may underlie appropriate expression of anxiety in face of the constantly changing environment.

## Results

### The aBLA and pBLA PNs have distinct projection patterns in dBNST

To investigate how aBLA and pBLA PNs connect to dBNST, we injected AAV vectors encoding EGFP under control of vGluT1 promotor (AAV-vGluT1-EYFP) into aBLA and AAV-vGluT1-mCherry into pBLA, which allowed us to anterogradely label their axonal terminals in dBNST (Fig. 1a, b). Three weeks later, we observed distinct projection patterns of the two groups of BLA PNs in dBNST (Fig. 1c, d). Specifically, the terminals of aBLA PNs were nearly exclusively localized within the adBNST, while the terminals of pBLA PNs were found to be packed within the ovBNST, with some scattered in adBNST (Fig. 1c).

To further confirm the distinct projections by the two BLA PN populations in dBNST, we conducted retrograde tracing experiments to label BLA neurons projecting to either the adBNST or ovBNST. Due to the small size of these two neighboring subregions, we only injected trace amount of retro-AAV-EF1a-mCherry (15 nl) into one of them to avoid diffusion to the other one. We then analyzed the distribution of two populations in BLA along the antero-posterior axis (Fig. 1e–l). Consistent with the findings from anterograde tracing, our analysis revealed that the BLA PNs projecting to adBNST were scattered throughout both the aBLA and pBLA subdivisions (Fig. 1g, h). On the other hand, those projecting to ovBNST were primarily localized in the pBLA with only few observed in aBLA (Fig. 1k, l).

Taken together, these findings consistently demonstrate distinct projection patterns of aBLA and pBLA PNs in dBNST, with the former mainly innervating the adBNST and the latter preferentially targeting the ovBNST over the adBNST.

### The aBLA and pBLA PNs have opposing effects on the activity of adBNST cells

We next explored how the projections from aBLA and pBLA PNs would affect the activity of their target cells in dBNST. To this end, we genetically expressed channelrhodopsin-2 (ChR2), a light-sensitive cation channel, in either of the two BLA PN populations. Four weeks later, we made whole-cell recordings of the adBNST or ovBNST neurons in the in vitro slices (Fig. 2a). The two dBNST cell populations were largely similar in their membrane properties except that the ovBNST neurons have far larger capacitance than their adBNST counterparts (Supplementary Fig. 1). We injected a small depolarization current (-15 pA) to evoke tonic firing of the recorded cells (-3 Hz). Then, we delivered a train of light pulses ($n = 20$, 10 Hz, 20 μW/mm$^2$) to optogenetically activate either the aBLA or pBLA inputs, and their effects on the firing of dBNST cells were examined (Fig. 2b–j).

We found that activation of both aBLA and pBLA inputs increased the firing frequency of their main target cells in adBNST (Fig. 2b–d) and ovBNST (Fig. 2e–g) respectively. The increase was more pronounced in the pBLA→ovBNST pathway, which aligns with the denser pBLA terminals in ovBNST. However, to our surprise, we found that pBLA inputs activation conspicuously suppressed, rather than augmented the firing of the adBNST cells (Fig. 2h–j).

### Chemogenetic activation of aBLA and pBLA PNs oppositely regulates the activity of adBNST cells in vivo

Given the glutamatergic and excitatory nature of the inputs from both aBLA and pBLA PNs, it is surprising to observe their opposing effects on the activity of their common target cells in adBNST. To investigate whether these opposite effects also occur in vivo, we expressed the excitatory designer receptor exclusively activated by designer drugs (DREADD) receptor, hM3Dq, in the aBLA or pBLA neurons projecting to dBNST. Additionally, GCaMP6s, a sensitive Ca$^{2+}$ indicator, was expressed in the adBNST cells (Fig. 2k). Mice were randomly administered with CNO and vehicle solution in a sequential manner, with a 4-day interval between administrations (Fig. 2k). The effectiveness of CNO administration to activate hM3Dq-expressing aBLA$^{→dBNST}$ and pBLA$^{→dBNST}$ PNs were confirmed by robust increase of the expression of c-Fos, a marker protein of neuronal activity, in these cells (Supplementary Fig. 2a–c). Calcium signals in adBNST cells were measured using fiber photometry before and after each administration. The pooled results showed that chemogenetic activation of the aBLA$^{→dBNST}$ and pBLA$^{→dBNST}$ PNs with CNO caused conspicuous increase and decrease, respectively, of the calcium signals in their target adBNST cells, compared to their respective vehicle controls (Fig. 2l–u; Supplementary Fig. 2d, e). These effects mirrored the observations made in vitro slices (Fig. 2d, j), indicating that the activation of the aBLA$^{→dBNST}$ and pBLA$^{→dBNST}$ PNs had opposing influences on the activity of their target adBNST cells.

### pBLA inputs activation evokes strong inhibition onto the adBNST cells

How then do the aBLA and pBLA inputs exert opposing effects on the activity of adBNST cells? Considering that external inputs typically alter the activity of target cells by shifting the balance between the excitatory/inhibitory (E/I) synaptic signals, we speculate that the contrasting effects of the aBLA and pBLA inputs may arise from their different effects on the E/I balance of the adBNST cells. To test this possibility, we directly measured the optogenetically-evoked excitatory and inhibitory postsynaptic currents (oEPSCs/oIPSCs) in these two pathways. We found that activation of either input elicited considerable oEPSCs and oIPSCs in adBNST cells (Fig. 3a). The monosynaptic nature of oEPSCs was confirmed by their complete blockage with TTX (1 μM), a sodium channel blocker and subsequent partial reversal by co-application of 4-AP, a potassium channel blocker (Supplementary Fig. 3b, c, h, i). On the other hand, the oIPSCs were found to be disynaptic, as their latency was nearly twice that of the oEPSCs (Supplementary Fig. 3d, e, j, k). Furthermore, the oIPSCs were fully blocked by either CNQX and AP5, the blockers of AMPA/NMDA receptors or PTX, a GABA$_A$ receptor antagonist, confirming their dependence on both glutamatergic and GABAergic transmission (Supplementary Fig. 3f, l).

Plotting the oEPSCs and oIPSCs amplitudes against the light intensity revealed that the aBLA-evoked oEPSCs in adBNST cells had higher amplitude relative to their oIPSC counterparts (Fig. 3b). By contrast, the pBLA-evoked oEPSCs had smaller amplitude relative to the oIPSCs (Fig. 3c). As a result, a significantly higher oIPSCs/oEPSCs amplitude ratio was found in the pBLA→adBNST pathway (Fig. 3d–f). Thus, pBLA activation caused a greater shift of the E/I balance toward inhibition in adBNST cells. Consistent with the dense pBLA innervation of ovBNST (Fig. 1a–d), pBLA activation evoked robust oEPSCs and oIPSCs in ovBNST cells (Supplementary Fig. 4a–d).

Lastly, we found that either partial or complete blockade of GABAergic transmission readily reversed the inhibitory effect of pBLA input activation on the activity of adBNST cells (Supplementary Fig. 5), confirming the critical role of strong GABAergic signals in pBLA-evoked suppression of adBNST cells.

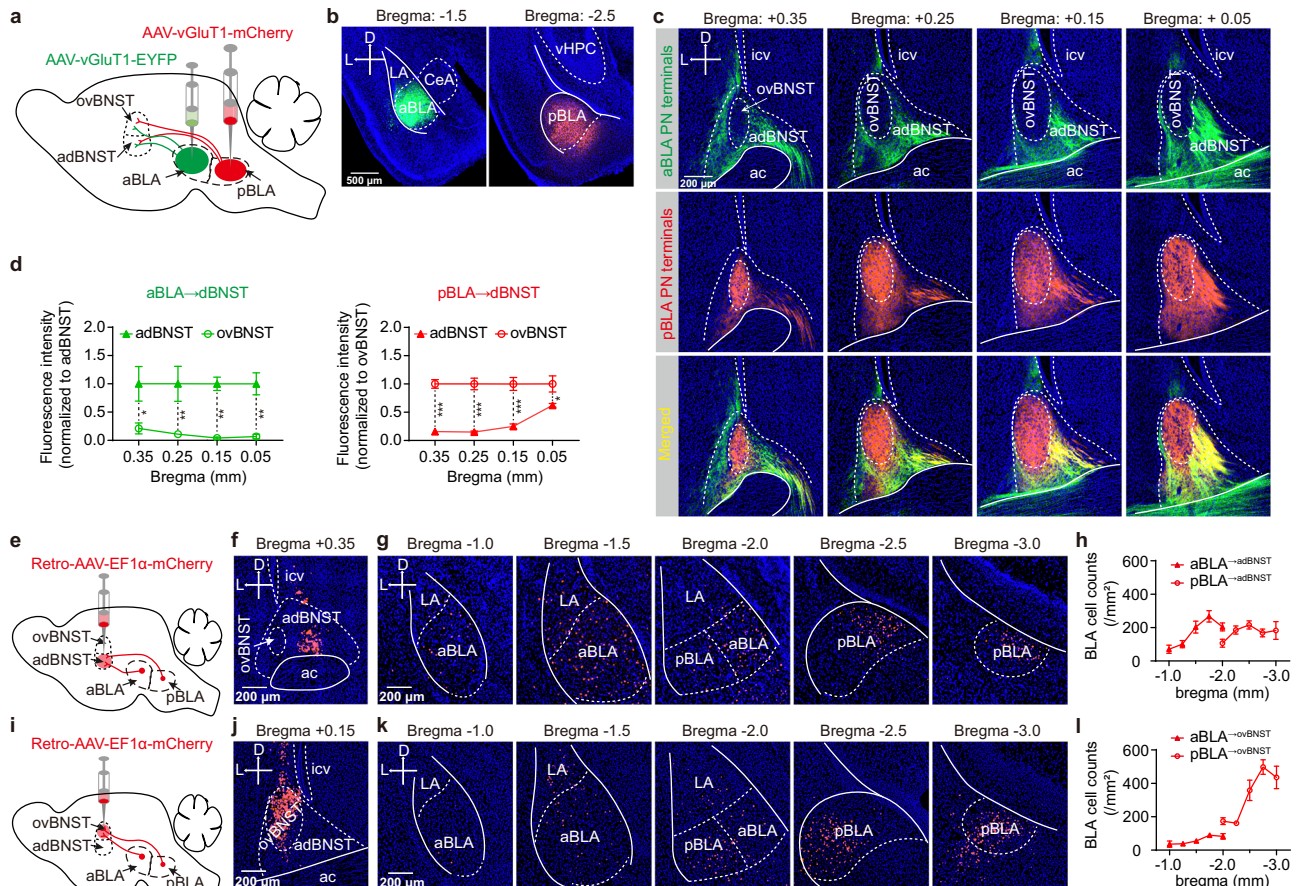

**Fig. 1 | The aBLA and pBLA PNs have distinct projection patterns in dBNST.** **a** Schematic showing injection of AAV vectors encoding EYFP and mCherry into aBLA and pBLA respectively, to anterogradely label the axonal terminals of their PNs in the two dBNST subregions. **b** Representative images showing EYFP and mCherry expression in aBLA (left) and pBLA (right) respectively. D dorsal, L left, CeA central amygdala, vHPC ventral hippocampus. This experiment was repeated 4 times with similar results. **c** Representative images showing the axonal terminals of aBLA (top) and pBLA (middle) PNs in the adBNST and ovBNST subregions of dBNST along the antero-posterior axis. Images were merged on the bottom. ac anterior commissure, icv intracerebroventricular. **d** Relative fluorescence intensity of the axonal terminals of aBLA and pBLA PNs in adBNST and ovBNST as shown in (**c**).

$n = 4$ mice, *$p < 0.05$, **$p < 0.01$, ***$p < 0.001$, two-way ANOVA with Bonferroni's multiple comparisons test. **e** Schematic showing injection of the retro-AAV vector encoding mCherry into adBNST to retrogradely label the BLA$^{→adBNST}$ PNs. **f** Representative image showing the injection site in adBNST. This experiment was repeated 4 times with similar results. **g** Representative images showing the retro-gradely labeled BLA$^{→adBNST}$ PNs along the antero-posterior axis. This experiment was repeated 4 times with similar results. **h** Quantification of the retrogradely labeled BLA$^{→adBNST}$ PNs in (**g**). $n = 4$ mice. **i–l** Same as (**e–h**) except that the retro-AAV was injected into the ovBNST. $n = 3$ mice. All data shown as means ± s.e.m. Source data including statistics are provided as a Source Date file.

## pBLA inputs activation drives strong feedforward inhibition from ovBNST to adBNST

We next proceeded to search for the sources of the pBLA-evoked GABAergic inhibition in adBNST cells. Given that pBLA PNs innervate both adBNST and ovBNST neurons which are primarily GABAergic[37,38], and, the ovBNST inhibitory neurons heavily target adBNST[25,39], we hypothesized that pBLA inputs could activate the local adBNST and/or the distal ovBNST interneurons to generate feedforward inhibition over the recorded adBNST cells. To distinguish between the contributions of local versus distal inhibition, we employed focalized light pulses (~120 μm in diameter) to sequentially activate the inhibitory network in either adBNST or ovBNST in a randomized order (Fig. 3g). We found that activation of the pBLA inputs in either subregion could readily elicit oIPSCs in adBNST cells (Fig. 3h). However, the oIPSCs evoked by activation in ovBNST were significantly larger than those by activation in adBNST (Fig. 3i). As a result, we concluded that pBLA recruits the inhibitory network in both adBNST and ovBNST to inhibit adBNST cells, with indirect inhibition from ovBNST playing a more prominent role.

The following observations further highlighted the essential role of ovBNST-to-adBNST inhibition. Firstly, when the pBLA inputs

in adBNST were selectively activated, it increased, rather than decreased, the firing of adBNST cells (Fig. 3j, k). This suggests that local adBNST inhibition is insufficient to counteract the direct excitatory effect of pBLA inputs. Secondly, when the pBLA inputs in ovBNST were activated, it consistently decreased the firing of adBNST cells (Fig. 3j, k). This implies that the indirect inhibition from ovBNST provides complementary, yet indispensable, support for the suppressive effect of pBLA inputs on the activity of adBNST neurons.

## aBLA→dBNST and pBLA→dBNST pathways oppositely regulate the anxiety-like behaviors in mice

The BLA→dBNST pathway is known to play a critical role in regulating negative moods such as anxiety[24–26]. Given the distinct projections of aBLA and pBLA PNs in dBNST and their opposing effects on altering the activity of adBNST cells, we set out to investigate whether they would also have different influences on anxiety. We selectively expressed ChR2 in aBLA or pBLA PNs and examined how optogenetic activation of their axonal terminals in dBNST would affect the mice's anxiety-like behaviors (Fig. 4). The results showed that activation of the aBLA$^{→dBNST}$ projections significantly increased the time mice spent in

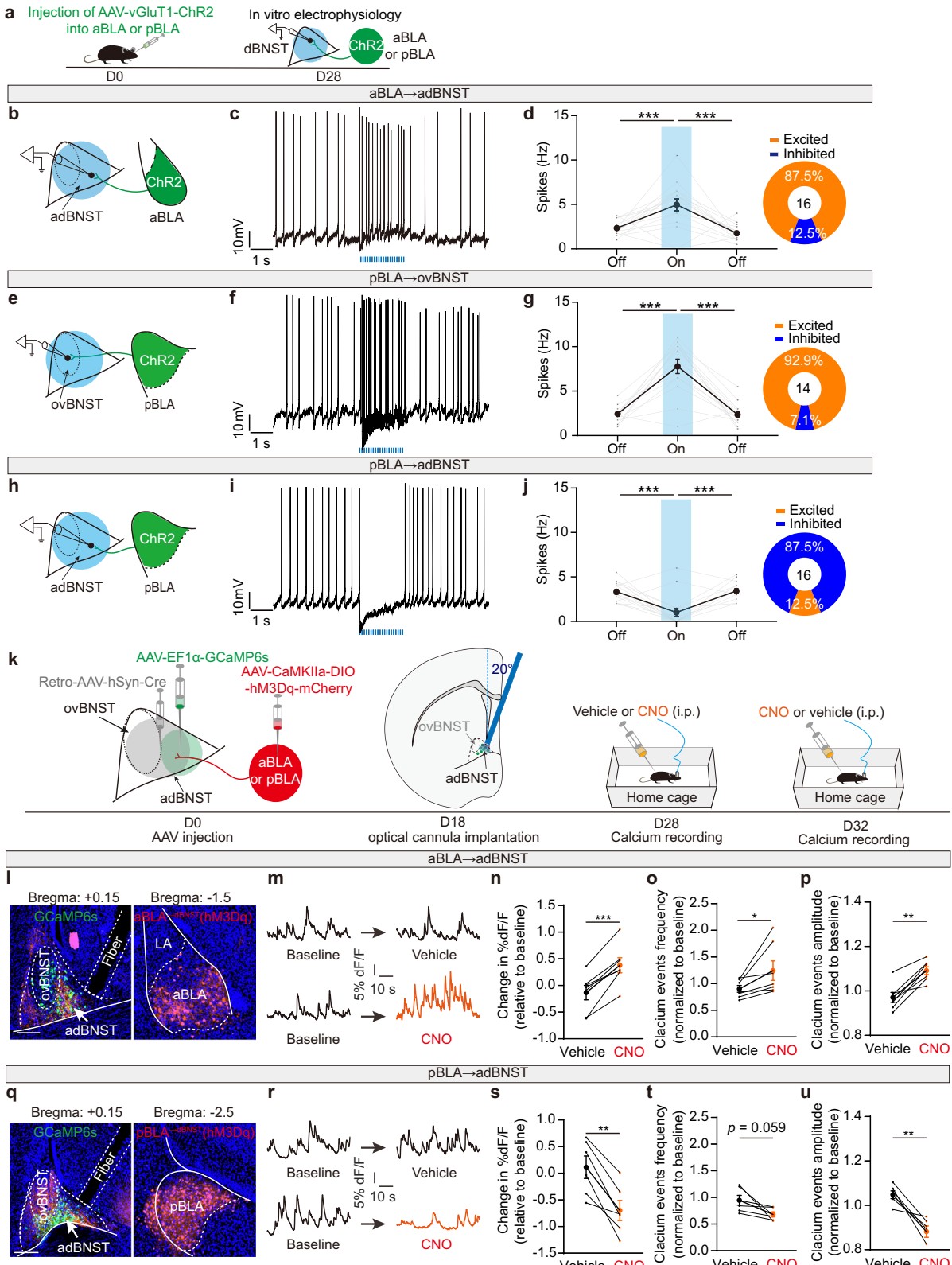

the central area of the open field test (OFT, Fig. 4a–e), as well as the time they spent in and their entries to the open arms in elevated plus maze (EPM, Fig. 4f, g). In contrast, optogenetic activation of the pBLA→dBNST projections had a clearly anxiogenic effect, as evidenced by less time mice spent in central region in the OFT and in the open arms of the EPM, and fewer entries to the open arms (Fig. 4h–n).

Optogenetic stimulation of the BLA projections to BNST also likely activates their collateral terminals targeting other regions through backpropagation of the evoked action potentials. To explore the possible involvement of these collateral projections, we delivered CNQX and AP5, the glutamate receptor antagonists or the vehicle solutions to dBNST 30 min prior to photoactivation of aBLA or pBLA

**Fig. 2 | aBLA and pBLA inputs oppositely regulate the activity of adBNST neurons. a** Experimental procedures for investigating the effect of optogenetic activation of aBLA or pBLA inputs on the dBNST neuronal activity in vitro. **b** Schematic showing recording of the firing of adBNST neurons to optogenetic activation of the entire aBLA inputs in dBNST. **c** Representative trace showing the firing of an adBNST neuron before, during and after aBLA inputs activation. **d** Summary of the spike frequency as shown in (**c**). Data from individual neurons were shown in gray. Inset showing the percentage of adBNST cells excited or inhibited following aBLA inputs activation. $n = 16$ neurons/5 mice; ***$p < 0.001$, one-way ANOVA with Bonferroni's multiple comparisons test. **e** Schematic showing recording of the firing of ovBNST neurons upon activation of pBLA inputs in dBNST. **f** Representative trace showing the firing of an ovBNST neuron before, during and after pBLA inputs activation. **g** Summary of the spike frequency as shown in (**f**). Data from individual neurons were shown in gray. Inset showing the percentage of ovBNST cells excited or inhibited following pBLA inputs activation. $n = 14$ neurons/5 mice, ***$p < 0.001$, one-way ANOVA with Bonferroni's multiple comparisons test. **h**–**j** Same as in (**e**–**g**) except that the recordings were made on the adBNST neurons. $n = 16$ neurons/5 mice, ***$p < 0.001$, one-way ANOVA with Bonferroni's multiple comparisons test. **k** Experimental procedures for investigating the effect of chemogenetic activation of aBLA$^{→dBNST}$ or pBLA$^{→dBNST}$ PNs on adBNST neuronal activity in vivo. i.p., intraperitoneal injection. **l** Left: Representative images showing the expression of GCaMP6s (green) in adBNST cells and the projections (red) from aBLA$^{→dBNST}$ PNs. The positioning of optical fiber was marked with the dashed lines. Right: Expression of hM3Dq in aBLA$^{→dBNST}$ PNs. Scale bar: 200 μm. This experiment was repeated 7 times with similar results. **m** Representative traces showing the calcium signals in adBNST cells before and after vehicle (top) or CNO (bottom) administration. **n**–**p** Summary of the calcium signal changes in adBNST cells following vehicle and CNO administration. Data from individual mice were shown in gray. $n = 7$ mice, *$p < 0.05$, **$p < 0.01$, ***$p < 0.001$, two tailed paired Student's $t$-test. **q**–**u** Same as in (**n**–**p**) except that hM3Dq was expressed in pBLA$^{→dBNST}$ PNs. $n = 6$ mice, **$p < 0.01$, two tailed paired Student's $t$-test. All data shown as means ± s.e.m. Source data including statistics are provided as a Source Date file.

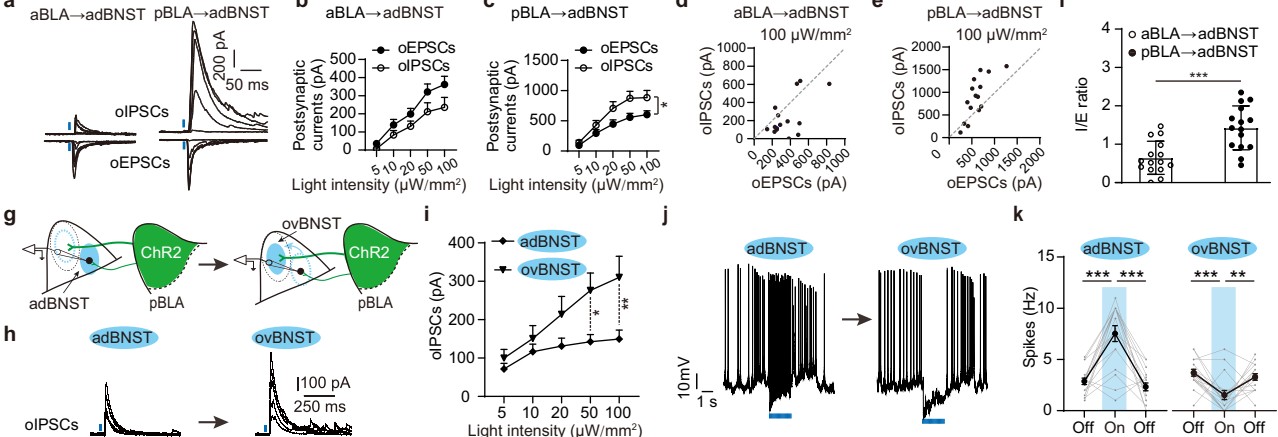

**Fig. 3 | pBLA inputs activation drives strong feedforward inhibition from ovBNST to adBNST. a** Representative traces showing oEPSCs and oIPSCs in adBNST neurons upon optogenetic activation of the entire aBLA (left) or pBLA (right) inputs with blue light pulses of increasing intensity. **b**, **c** Summary of the oEPSCs and oIPSCs amplitudes against the increasing intensity of light stimuli delivered to aBLA (**b**) and pBLA (**c**) inputs. $n = 15$ neurons/5 mice for both groups. *$p < 0.05$, two-way ANOVA with Bonferroni's multiple comparisons test. **d**, **e** Distribution of the oEPSCs and oIPSCs amplitudes in each recorded adBNST cells when the light stimuli were delivered to aBLA (**d**) and pBLA (**e**) inputs at an intensity of 100 μW/mm². **f** Summary of the I/E ratio in (**d** and **e**). $n = 15$ neurons/5 mice for both groups. ***$p < 0.001$, two tailed unpaired Student's $t$-test. **g** Schematic diagrams showing recording of the oIPSCs in adBNST neurons or their firings upon selective activation of the pBLA terminals in either ovBNST or adBNST with focalized light stimuli. Note that the stimuli were sequentially delivered to either of the two subregions at a randomized order. **h** Representative traces showing oIPSCs recorded from one identical adBNST neuron when the light stimuli were delivered to adBNST (left) and ovBNST (right). **i** Summary of the oIPSCs amplitudes in adBNST neurons as shown in (**h**). $n = 14$ neurons/4 mice. *$p < 0.05$, **$p < 0.01$, two-way ANOVA with Bonferroni's multiple comparisons test. **j** Representative traces showing the firing of one identical adBNST cell in response to the light activation of pBLA terminals in adBNST (left) and ovBNST (right). **k** Summary of the spike frequency of adBNST neurons before, during and after selective activation of pBLA inputs in either the adBNST or ovBNST as shown in (**j**). $n = 17$ neurons/5 mice; **$p < 0.01$, ***$p < 0.001$, one-way ANOVA with Bonferroni's multiple comparisons test. All data shown as means ± s.e.m. Source data including statistics are provided as a Source Date file.

terminals[23,31]. We found that pretreatment with CNQX and AP5 but not vehicle solution deprived the modulatory effects of both BLA→dBNST pathways in regulating the anxiety-like behaviors in both OFT and EPM (Supplementary Fig. 6). These findings suggest that the direct aBLA and pBLA inputs onto dBNST are required for the anxiety regulation of BLA→dBNST pathways. The roles of their collateral projections, if there are some, are insufficient to mediate the anxiety-modulatory effects. It should be mentioned that, CNQX and AP5 infusion into dBNST was found to be anxiolytic in single-housed animals with high anxiety[25], while such treatment failed to influence the anxiety-like behaviors in group-housed animals with relative low anxiety in our study (Supplementary Fig. 6).

Next, by selectively expressing NpHR, a light-sensitive chloride ion channel in either aBLA or pBLA PNs, we observed that optogenetic inhibition of aBLA→adBNST pathway elicited an anxiogenic response, whereas inhibition of pBLA→dBNST circuit exhibited an anxiolytic effect (Supplementary Fig. 7).

Therefore, our findings suggest that the aBLA→dBNST and pBLA→dBNST pathways have opposing effects on the anxiety-like behaviors in mice.

### lpBLA→ovBNST and mpBLA→adBNST pathways oppositely regulate the anxiety-like behaviors in mice

Having observed that activation of the pBLA fibers in ovBNST and adBNST also oppositely regulate the adBNST cells (Fig. 3), we next sought to determine the specific roles of pBLA→ovBNST and pBLA→adBNST subcircuits in anxiety regulation. Before delving into this, we first determined whether the pBLA terminals in ovBNST and adBNST were from the same or different PN populations by using retrobeads-assisted retrograde labeling of the pBLA$^{→ovBNST}$ and pBLA$^{→adBNST}$ projectors (Supplementary Fig. 8a). We found that the two projectors were clearly separate in pBLA, with the former mainly located in the lateral part and the latter in the medial part (Supplementary Fig. 8b, c). This segregation was further validated by the

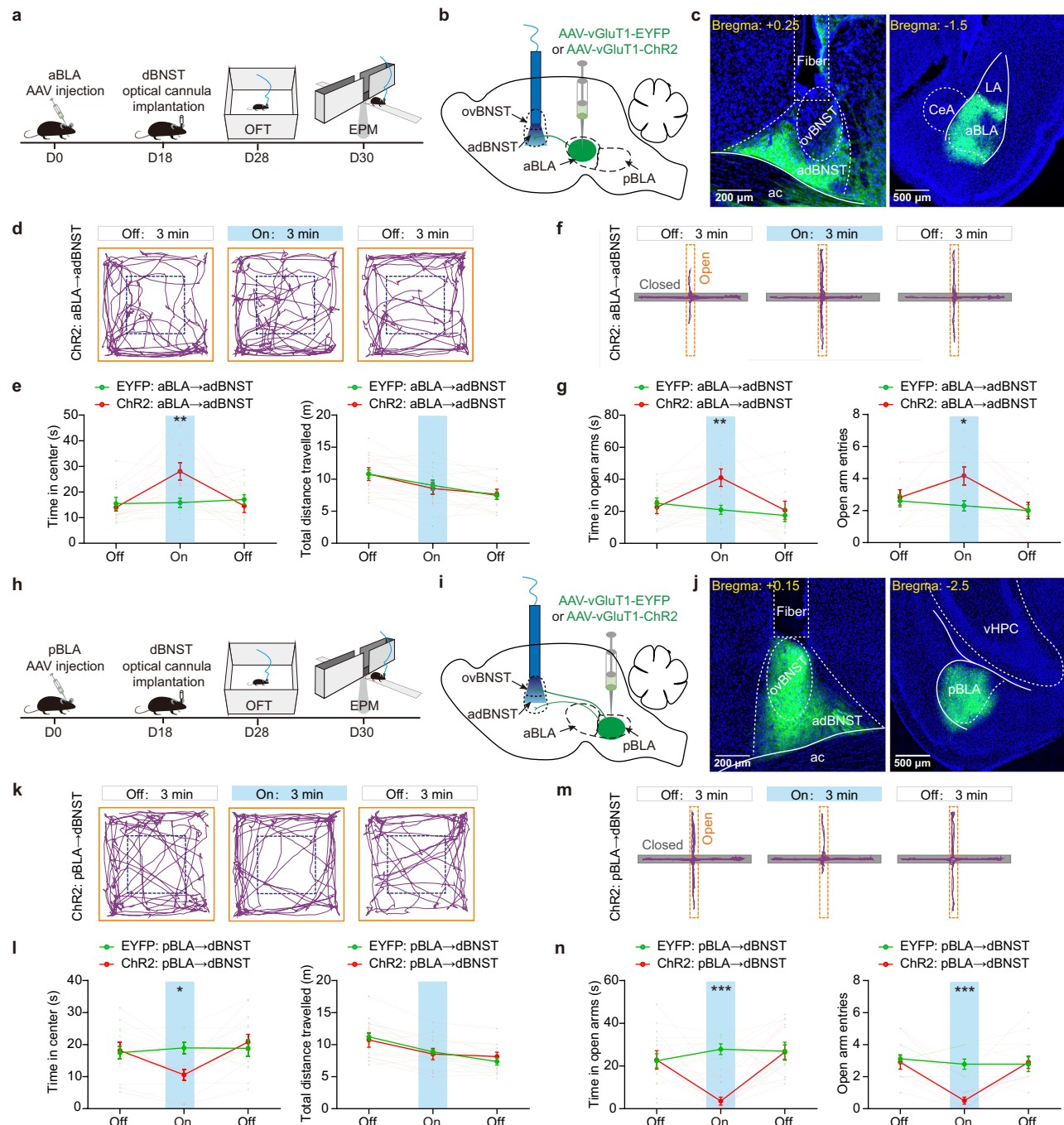

**Fig. 4 | Optogenetic activation of aBLA→dBNST and pBLA→dBNST pathways oppositely regulates the anxiety-like behaviors in mice. a**, Experimental procedures for investigating the effect of optogenetic activation of aBLA→dBNST pathway on mice's anxiety-like behaviors. OFT, open field test, EPM, elevated plus maze. **b**, Schematic showing injection of AAV vectors encoding EYFP or ChR2 into aBLA and implantation of optical cannula onto dBNST for optogenetic manipulations. **c**, Left: Representative image showing the axonal terminals from aBLA PNs in dBNST. The positioning of optical fiber was marked with the dashed lines. Right: ChR2 expression in aBLA PNs. This experiment was repeated 11 times with similar results. **d**, Representative activity tracking across epochs in OFT for a mouse expressing ChR2 in aBLA. The designated center zone was marked with dashed lines. **e**, Summary of the time in center and total distance travelled before, during

and after light stimulation of the EYFP- or ChR2-expressing aBLA inputs. Data from individual mice were shown in light colors. $n = 10$ (EYFP), 11 (ChR2) mice. $**p < 0.01$, two-way ANOVA with Bonferroni's multiple comparisons test. **f**, Representative activity tracking across epochs in EPM for a mouse expressing ChR2 in aBLA. **g**, Summary of the time in open arms and open-arm entries measured before, during and after light stimulation of the EYFP- or ChR2-expressing aBLA inputs. Data from individual mice were shown in light colors. $*p < 0.05, **p < 0.01$, two-way ANOVA with Bonferroni's multiple comparisons test. **h–n**, same as in (**a–g**), except that EYFP or ChR2 was expressed in the pBLA PNs. $n = 9$ (EYFP), 10 (ChR2) mice. $*p < 0.05, ***p < 0.001$, two-way ANOVA with Bonferroni's multiple comparisons test. All data shown as means ± s.e.m. Source data including statistics are provided as a Source Date file.

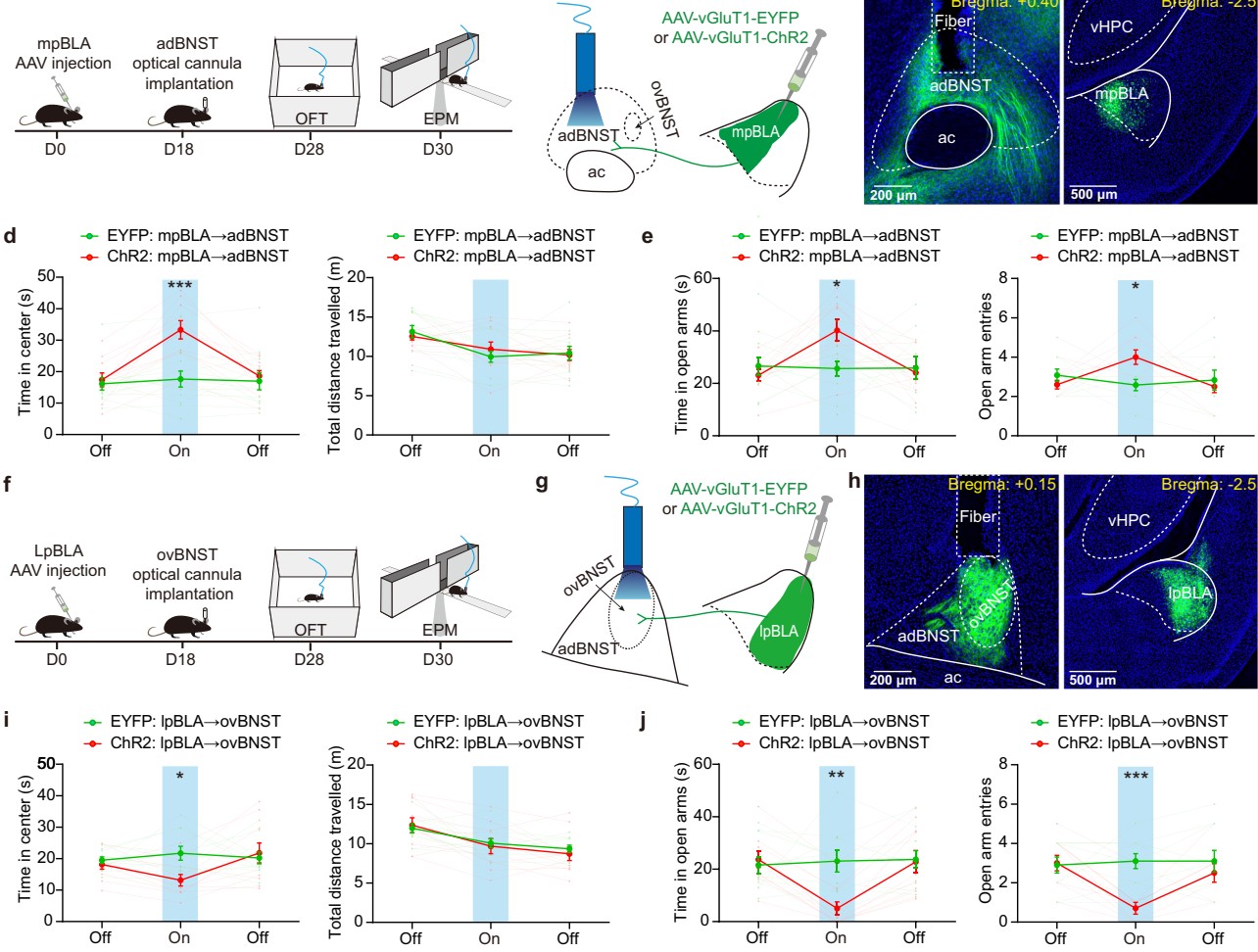

**Fig. 5 | Optogenetic activation of mpBLA→adBNST and lpBLA→ovBNST pathways oppositely regulates the anxiety-like behaviors in mice. a** Experimental procedures for investigating the effect of optogenetic activation of mpBLA→adBNST pathway in regulating the mice's anxiety-like behaviors. OFT, open field test, EPM, elevated plus maze. **b** Schematic showing injection of AAV vectors encoding EYFP or ChR2 into mpBLA and implantation of cannula onto adBNST for optogenetic manipulations. **c** Left: Representative image showing the axonal terminals from mpBLA PNs. The position of optical fiber was marked with the dashed lines. Right: ChR2 expression in mpBLA PNs. This experiment was repeated 10 times with similar results. **d** Summary of the time in center region and total distance travelled in OFT before, during and after light stimulation of the

EYFP- or ChR2-expressing mpBLA projections. $n = 12$ (EYFP), 10 (ChR2) mice. Data from individual mice were shown in gray. ***$p < 0.001$, two-way ANOVA with Bonferroni's multiple comparisons test. **e** Summary of the time in open arms and open-arm entries in EPM before, during and after light stimulation of EYFP- or ChR2-expressing mpBLA projections. $n = 12$ (EYFP), 10 (ChR2) mice. *$p < 0.05$, two-way ANOVA with Bonferroni's multiple comparisons test. **f–j** same as in (**a–e**), except that EYFP or ChR2 was expressed in lpBLA PNs and the cannula was implanted onto ovBNST. $n = 10$ mice for both. *$p < 0.05$, **$p < 0.01$, ***$p < 0.001$, two-way ANOVA with Bonferroni's multiple comparisons test. All data shown as means ± s.e.m. Source data including statistics are provided as a Source Date file.

anterograde tracing experiment. To reduce diffusion, a small volume (30 nl) of the virus was injected into either the lateral or medial parts of pBLA. Consistent with the retrograde labeling result, the lpBLA PNs primarily sent their fibers to ovBNST and mpBLA PNs to adBNST (Supplementary Fig. 8d–i). Therefore, the pBLA PNs targeting the ovBNST and adBNST mainly represent two spatially separate populations.

Next, we expressed ChR2 in either the mpBLA or lpBLA PNs and optogenetically activated their terminals in adBNST and ovBNST, respectively, to determine the specific roles of mpBLA→adBNST vs lpBLA→ovBNST pathways in anxiety regulation. Our data demonstrate that the two circuits made by the mpBLA and lpBLA PNs onto dBNST have opposing roles in anxiety regulation, with the former being anxiolytic and the latter anxiogenic (Fig. 5).

Based on the behavioral findings shown in Figs. 4 and 5, we speculate that for the complex circuits made by the distinct BLA PNs along the antero-posterior axis onto dBNST, it is their exact targets

(ovBNST or adBNST) that determine their respective effects on anxiety regulation (anxiolytic or anxiogenic).

## The aBLA→adBNST and lpBLA→ovBNST pathways oppositely regulate the adBNST neuronal activity and the mice's anxiety-like behaviors in OFT and EPM

Our aforementioned findings have revealed two distinct BLA→dBNST pathways which act to oppositely regulate the activity of adBNST neurons in the mice's home cage and their anxiety-like behaviors in OFT and EPM. To explore how the two pathways would simultaneously regulate the adBNST neuronal activity as well as the mice's behaviors in OFT and EPM, we selectively expressed ChR2 in either the aBLA or lpBLA neurons projecting to dBNST. Since both ChR2 and GCaMP6 were activated by blue light, we used jRGECO1a which is activated by yellow light, as an alternative Ca²⁺-sensitive indicator, and expressed it in the adBNST cells (Fig. 6a, b, i). We observed that photoactivation of aBLA$^{→adBNST}$ PNs yielded increase of

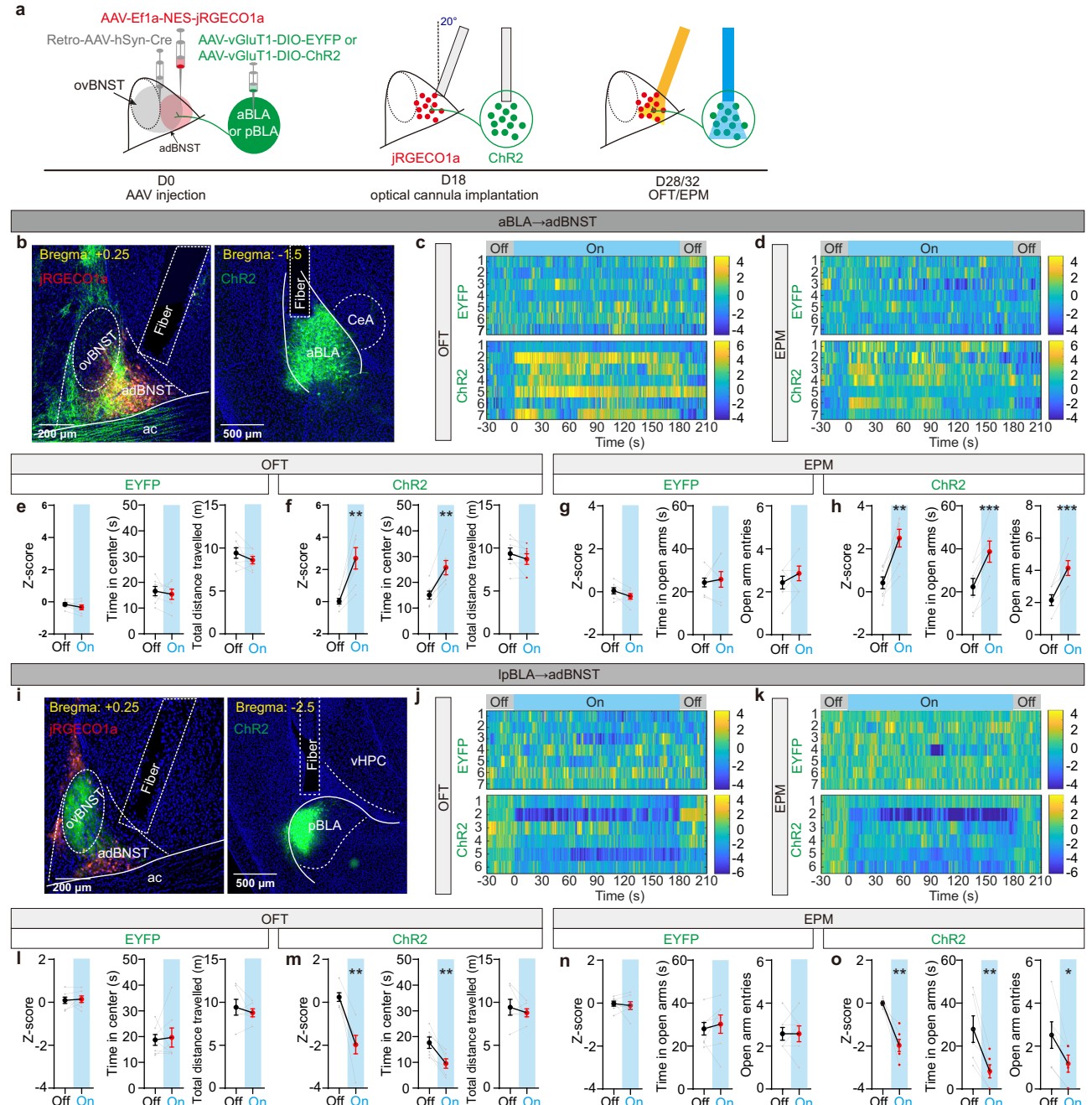

**Fig. 6 | The aBLA→adBNST and lpBLA→ovBNST pathways oppositely regulate the adBNST neuronal activity and anxiety-like behaviors in OFT and EPM.**
**a** Experimental procedures for investigating the effect of optogenetic activation of aBLA$^{→dBNST}$ or pBLA$^{→dBNST}$ PNs on adBNST neuronal activity and anxiety-like behaviors in OFT and EPM. OFT, open field test, EPM, elevated plus maze. **b** Left: Representative images showing the expression of jRGECO1a (red) in adBNST cells and the projections (green) from aBLA$^{→dBNST}$ PNs. Right: Expression of ChR2 in aBLA$^{→dBNST}$ PNs. The positions of optical fiber were marked with the dashed lines. This experiment was repeated 7 times with similar results. **c**, **d** Heat maps showing the calcium signal in the adBNST neurons of individual mice before, during and after light stimulation of the EYFP- (top) or ChR2-expressing (bottom) aBLA$^{→dBNST}$

PNs in OFT (**c**) and EPM (**d**). $n = 7$ mice for both groups. **e**, **f** Summary of the calcium signal, time in center and total distance travelled in OFT before and during light stimulation of the EYFP- (**e**) or ChR2-expressing (**f**) aBLA$^{→dBNST}$ PNs. Data from individual mice were shown in gray. **$P < 0.01$, two tailed paired Student's $t$-test. **g**, **h** Summary of the calcium signal, time in open arms and open arm entries in EPM before and during light stimulation of the EYFP- (**g**) or ChR2-expressing (**h**) aBLA$^{→dBNST}$ PNs. Data from individual mice were shown in gray. **$P < 0.01$ and ***$P < 0.001$, two tailed paired Student's $t$-test. **i**–**o** Same as in (**b**–**h**) except that ChR2 was expressed in lpBLA$^{→dBNST}$ PNs. $n = 7$ (EYFP), 6 (ChR2) mice. *$P < 0.05$ and **$P < 0.01$, two tailed paired Student's $t$-test. All data shown as means ± s.e.m. Source data including statistics are provided as a Source Date file.

the activity of adBNST neurons with concomitant reduction of the anxiety-like behaviors in both the OFT and EPM (Fig. 6c–h). In contrast, photoactivation of lpBLA$^{→ovBNST}$ PNs reduced the activity of adBNST neurons, but increased the anxiety-like behaviors in both the OFT and EPM (Fig. 6j–o).

**The aBLA→adBNST and lpBLA→ovBNST pathways synergistically encode the anxiety-relevant cues in the changing environment**
Next, we investigated how these two pathways respond to anxiety-relevant cues in the environment. We expressed GCaMP6s in either aBLA or lpBLA PNs and used fiber photometry to monitor the real-

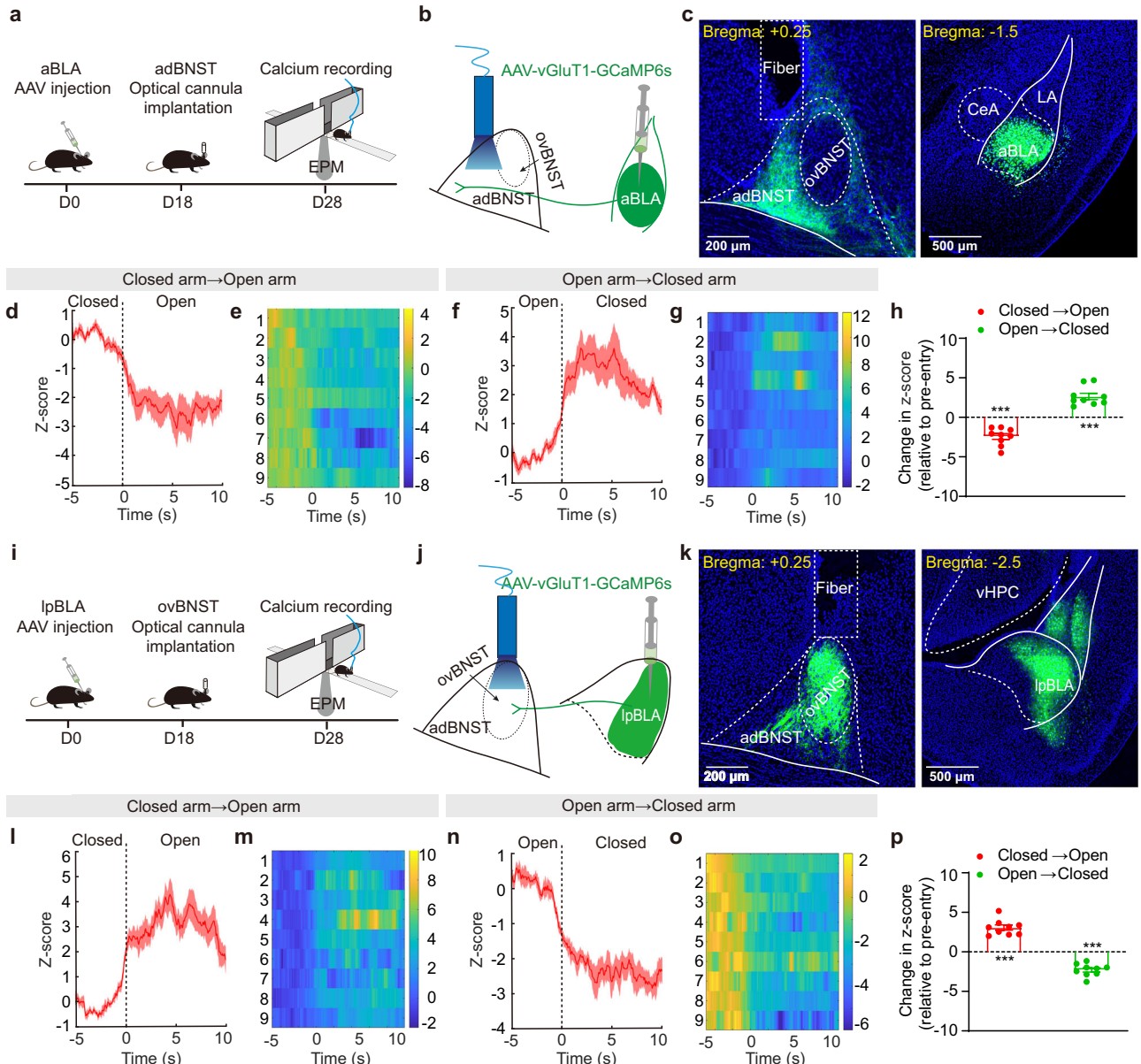

**Fig. 7 | The aBLA→adBNST and lpBLA→ovBNST pathways synergistically encode the anxiety-relevant cues in the environment. a** Experimental procedures for real-time monitoring of the calcium signal changes in the aBLA→adBNST axonal terminals when the mice explored in the elevated plus maze (EPM). **b** Schematic showing injection of AAV vectors encoding GCaMP6s into aBLA and implantation of the recorded optical cannula onto adBNST. **c** Left: Representative image showing the axonal terminals expressing GCaMP6s from aBLA PNs. The positioning of optical cannula was marked with the dashed lines. Right: GCaMP6s expression in aBLA PNs. This experiment was repeated 9 times with similar results. **d** Changes of average calcium transients in aBLA→adBNST fibers when the mice moved from the closed arm to the open arm. $n = 9$ mice. **e** Heat maps showing the calcium fluorescence in the aBLA→adBNST terminals of individual mice (2–7 trails for each) in (**d**). **f, g** Same as in (**d–e**) except that the calcium signals were measured when the mice transitioned from open to the closed arm. **h** summary of the changes of the average calcium signals in aBLA→adBNST fibers when mice explored between the closed and open arms of EPM. ***$p < 0.001$, two tailed one sample $t$ test. **i–p** Same as in (**a–h**), except that GCaMP6s was expressed in lpBLA PNs and the cannula was implanted onto ovBNST. $n = 9$ mice. ***$p < 0.001$, two tailed one sample $t$-test. All data shown as means ± s.e.m. Source data including statistics are provided as a Source Date file.

time calcium activity of their axonal terminals in adBNST and ovBNST, respectively, as mice explored the EPM (Fig. 7, Supplementary Fig. 9–10). As illustrated in Fig. 7d, e, h, the activity of aBLA→adBNST terminals exhibited a significant decrease when mice transitioned from the "anxiolytic" closed arm to the "anxiogenic" open arm, and an increase when they returned to the closed arm (Fig. 7f, g, h). In contrast, the activity of lpBLA→ovBNST terminals significantly increased when mice moved from the closed to open arm, and decreased when they returned to the closed arm (Fig. 7i–p). Thus, both the aBLA→adBNST and lpBLA→ovBNST pathways are recruited to encode the anxiolytic and anxiogenic cues in the environment, with their activity showing temporally synchronized but diametrically opposite changes.

## The aBLA→adBNST and lpBLA→ovBNST PNs are reciprocally inhibited through recruiting the local interneurons

How do the two BLA→dBNST circuits exhibit opposite responses when mice explore between the open and closed arms of the EPM? Previous studies have shown that reciprocal inhibition exists between the aBLA and pBLA excitatory neural populations via engaging local inhibitory networks[27]. If such reciprocal inhibition also occurs between the two specific dBNST-projecting populations in aBLA and lpBLA, it could at

least partially explain the aforementioned observation. To explore this possibility, we used viral approaches to express mCherry in both dBNST-projecting populations in BLA, with ChR2 expressed only in one of them (Fig. 8a, b, j, k). Subsequently, we optogenetically activated the ChR2-expressing PNs in one subregion and examined their impact on the firing of mCherry-expressing-only PNs in the other.

Our findings demonstrated that activating aBLA[→adBNST] PNs significantly reduced the firing frequency of lpBLA[→ovBNST] PNs, indicating inhibition of aBLA[→adBNST] PNs over their lpBLA[→ovBNST] counterparts (Fig. 8c–e). This inhibition appears to be mediated by the recruitment of local interneurons. Firstly, optogenetic stimulation of aBLA[→adBNST] PNs markedly increased the firing of interneurons in lpBLA (Fig. 8f–h), which are easily distinguishable from the PNs based on their small size, low capacitance and firing properties[40–42] (Supplementary Fig. 11a–c). Secondly, the observed inhibition was abolished by 100 μM PTX, a GABA$_A$ receptor antagonist (Fig. 8i). Similarly, when investigating the effect of optogenetic activation of lpBLA[→ovBNST] PNs on the firing of aBLA[→adBNST] PNs, we observed robust inhibition that was also dependent on GABA$_A$ receptors (Fig. 8j–r). These results strongly suggest the presence of significant reciprocal inhibition between the two BLA PN populations that terminate on distinct dBNST subregions. This mutual inhibition may, at least in part, contribute to the opposite responses of the aBLA→adBNST and lpBLA→ovBNST pathways when mice shuttle between the closed and open arms in EPM.

### Whole-brain mapping of the monosynaptic inputs to aBLA[→adBNST] and lpBLA[→ovBNST] PNs

Alternatively, the opposing responses of the two BLA[→dBNST] PN populations to anxiety-relevant cues may be attributed to their different afferent inputs. To explore this possibility, we employed projection-specific retrograde transsynaptic tracing to map brain-wide monosynaptic inputs to these two populations[43] (Fig. 9a–d). As shown in Fig. 9c, d, both aBLA[→adBNST] and lpBLA[→ovBNST] PNs received monosynaptic inputs from multiple brain areas associated with diverse functions, including sensory, integrative, contextual, and neuromodulatory processes[44–47]. This indicates that both types of BLA[→dBNST] PNs are capable of integrating diverse information from various upstream regions. Notably, the afferents to the two BLA PN populations differ significantly in many upstream brain regions, including auditory areas (AUD), agranular insular area (AI), ventral CA1 (CA1v) and entorhinal area (ENT) (Fig. 9c).

## Discussion

In this study, by conducting comprehensive investigation into the architecture and function of the BLA→dBNST circuits, we have made three major findings. First, we observed the existence of two anatomically adjacent outputs from separate groups of BLA PNs that target adBNST and ovBNST respectively. While aBLA PNs primarily terminate in adBNST with their activation exciting the adBNST neurons and inducing anxiolysis, pBLA PNs, particularly those in lpBLA, project heavily to ovBNST with their activation suppressing the adBNST neurons and causing anxiogenesis. Second, we found that the two pathways respond cohesively to changes in the environment, as demonstrated by their synchronized yet opposing responses to the anxiolytic and anxiogenic cues in EPM paradigm. Last, we observed the coordination between the two pathways is attributed, if not wholly, to the reciprocal suppression between the aBLA[→adBNST] and lpBLA[→ovBNST] PNs in BLA, the different inputs they receive, as well as the interaction of their target cells in dBNST. These findings provide valuable insights into how functionally distinct circuits get effectively coordinated to enable appropriate mood and behavioral responses to the ever-changing environment (Fig. 10).

Considerable heterogeneity has been identified among individual BLA PNs along the antero-posterior axis in terms of their gene expression, cell morphology and function[13,14,27,48,49]. Furthermore, these BLA PNs display notable differences in their axonal projections, as they terminate in various brain regions or even different subdivisions within a region[27,50,51]. For instance, BLA PN populations targeting specific subdivisions of the central amygdala (CeA), NAc, and ventral hippocampus (vHPC) were observed to have distinct distribution patterns[27,29,31,50]. Nevertheless, previous studies have consistently demonstrated that BLA PNs targeting the dBNST primarily project their axonal terminals to the adBNST, regardless of their aBLA or pBLA origin, with only a few pBLA fibers seen in ovBNST[50,51]. In the present study, by using both anterograde and retrograde tracing approaches, we found that aBLA PNs almost exclusively project to the adBNST, which aligns with previous findings[25,50]. In contrast, pBLA PNs send axonal terminals to both the adBNST and ovBNST, with higher density observed in the ovBNST compared to the adBNST. Further analysis demonstrated separate distributions of the pBLA[→adBNST] and pBLA[→ovBNST] PNs in pBLA, with the former primarily situated in the medial region and the latter in lateral region. These findings strongly suggest that the BLA PNs innervating different subregions of dBNST also exhibit distinct distribution along the antero-posterior axis. One explanation for the previous lack of observation of dense BLA fibers in the ovBNST could potentially be attributed to the bias in the injection of tracing viral vectors towards the anterior edge of the pBLA (~ −2.0 mm from bregma)[44,50,52] as opposed to the more posterior (−2.5 mm from bregma) region investigated in the current study, which allows for the exploration of the majority of pBLA[→ovBNST] PNs (Fig. 1).

Functionally, the activity of aBLA and pBLA neuronal populations exhibits opposite correlations with emotions[27]. However, the distinct PN subsets in both subdivisions of BLA exert different influences on emotions depending on their connectivity[31,53–57]. For example, in the aBLA, activating PNs projecting to the vHPC increased anxiety[31]. In contrast, our findings demonstrate that activating the aBLA[→adBNST] projections produces a clear anxiolytic effect, which is consistent with previous studies[24–26]. Similarly, while unbiased activation of the entire pBLA[→dBNST] PN projections increases anxiety (Fig. 4), selective activation of the mpBLA and lpBLA inputs to dBNST yields distinct effects: mpBLA exerts an anxiolytic effect, whereas lpBLA exerts an anxiogenic effect (Fig. 5). These opposing influences may arise from the different targets of these PNs in the dBNST, with the ovBNST (the target of the lpBLA PNs) facilitating anxiety and the adBNST (the target of the mpBLA PNs) suppressing anxiety[25,34–36].

Although the observation that BLA sends dense projections to dBNST has been known for decades[51], its impact on the activity and function of dBNST cells remains understudied. In this study, we conducted comprehensive investigations on the projection patterns of PNs from aBLA and pBLA to dBNST, as well as their regulatory effects on the dBNST cells. We found that while PNs from aBLA and pBLA exhibit distinct projection patterns in the dBNST, they shared the same projection targets in adBNST. Surprisingly, our investigations using both in vitro slice preparations and in vivo animal models consistently revealed that the activation of these PNs had opposing influences on the activity of adBNST cells (Figs. 2, 6). Specifically, the activation of aBLA PNs increased adBNST cell firings, while the activation of pBLA PNs suppressed it by recruiting strong feedforward inhibition from ovBNST onto adBNST (Fig. 3). These findings suggest that among the dense projections from BLA to dBNST, some directly activate the adBNST cells[25], while others terminating in the ovBNST serve as a brake mechanism to prevent excessive activation of adBNST cells by BLA PNs.

A similar braking mechanism has also been found in BLA projections to other regions such as CeA[22,27,29]. The BLA PNs innervate the medial part of CeA (CeM), the primary output subregion of CeA, and the capsular/lateral subregion of CeA (CeC/L) which is known as an inhibitory "relay station" in CeA[22]. Consistently, it has been observed that activation of BLA PNs terminals in CeC/L leads to a net inhibition in CeM cells, primarily due to the robust feedforward inhibition from

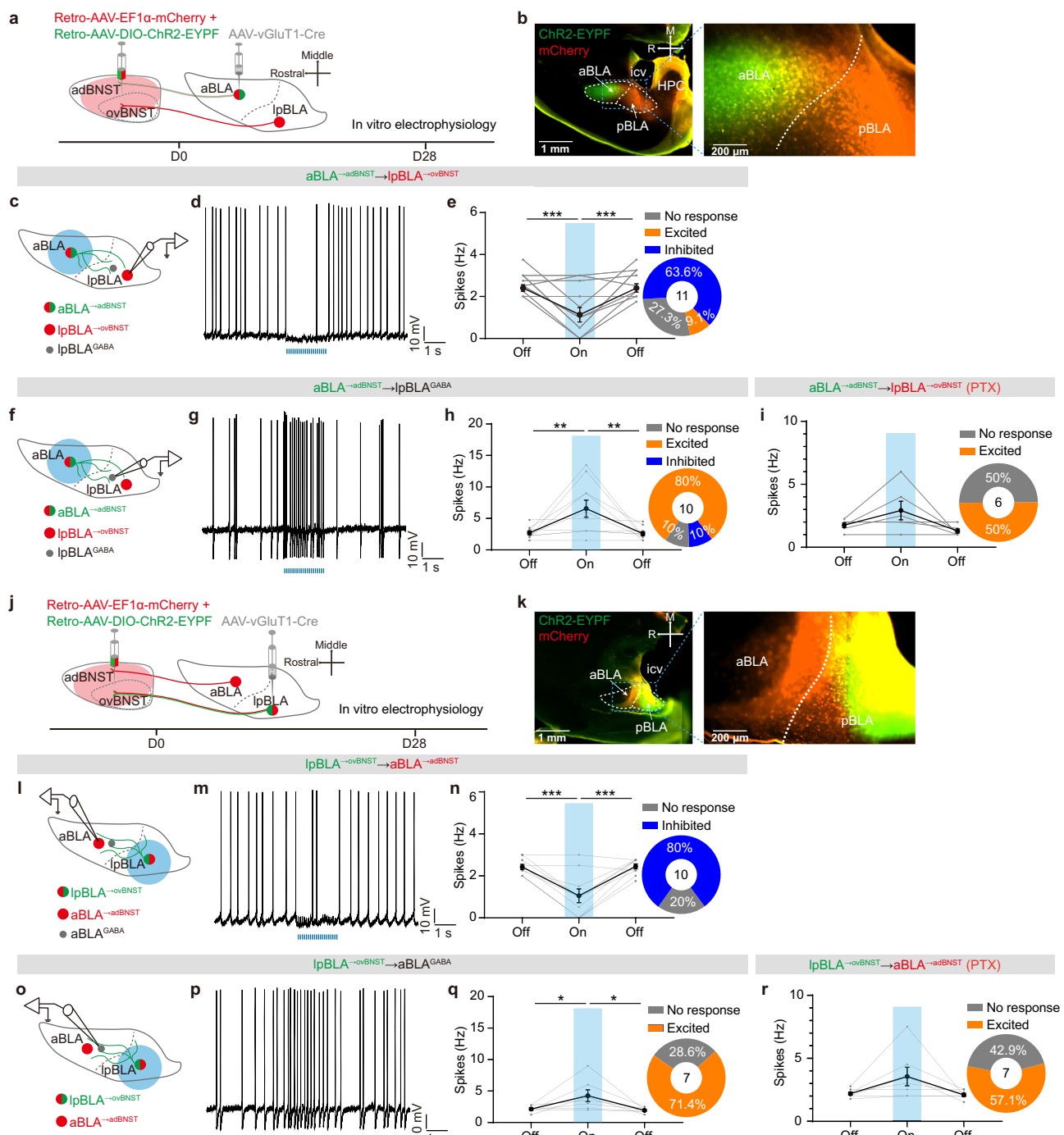

**Fig. 8 | The aBLA[→adBNST] and lpBLA[→ovBNST] PNs are reciprocally inhibited through recruiting the local interneurons. a** Experimental procedures for investigating the effect of optogenetic activation of aBLA[→adBNST] PNs on the activity of lpBLA[→ovBNST] PNs. **b** Left: Representative image showing unbiased mCherry expression in both aBLA[→adBNST] and pBLA[→adBNST] PNs and selective ChR2 expression in aBLA[→adBNST] PNs in a horizontal slice. Right: The magnified image of BLA. icv intracerebroventricular; HPC hippocampus. This experiment was repeated 4 times with similar results. **c** Schematic showing recording of the firing of lpBLA[→ovBNST] PNs upon optogenetic activation of aBLA[→adBNST] PNs. **d** Representative trace showing the firing of lpBLA[→ovBNST] PN before, during and after activation of aBLA[→adBNST] PNs. **e** Summary of the spike frequency as shown in (**d**). Data from individual neurons were shown in gray. Inset showing the percentage of lpBLA[→ovBNST] PNs excited or inhibited following aBLA[→adBNST] neuronal activation. $n = 11$ neurons/4 mice; ***$p < 0.001$, one-way ANOVA with Bonferroni's multiple comparisons test. **f–h** Same as in (**c–e**), except

that the GABAergic interneurons in lpBLA (lpBLA[GABA]) were recorded. $n = 10$ neurons/4 mice; **$p < 0.01$, one-way ANOVA with Bonferroni's multiple comparisons test. **i** Summary of the spike frequency in lpBLA[→ovBNST] PNs before, during and after aBLA[→adBNST] neuronal activation in the presence of PTX (100 μM). Data from individual neurons were shown in gray. Inset showing the percentage of lpBLA[→ovBNST] PNs excited or inhibited following aBLA[→adBNST] neuronal activation. $n = 6$ neurons/3 mice. **j–r** Same as in (**a–i**) except that ChR2 was selectively expressed in lpBLA[→ovBNST] PNs and the recordings were made on the aBLA neurons. **k** This experiment was repeated 4 times with similar results. **n** $n = 10$ neurons/4 mice; ***$p < 0.001$, one-way ANOVA with Bonferroni's multiple comparisons test. **q** $n = 7$ neurons/3 mice; *$p < 0.05$, one-way ANOVA with Bonferroni's multiple comparisons test. **r** $n = 7$ neurons/3 mice. All data shown as means ± s.e.m. Source data including statistics are provided as a Source Date file.

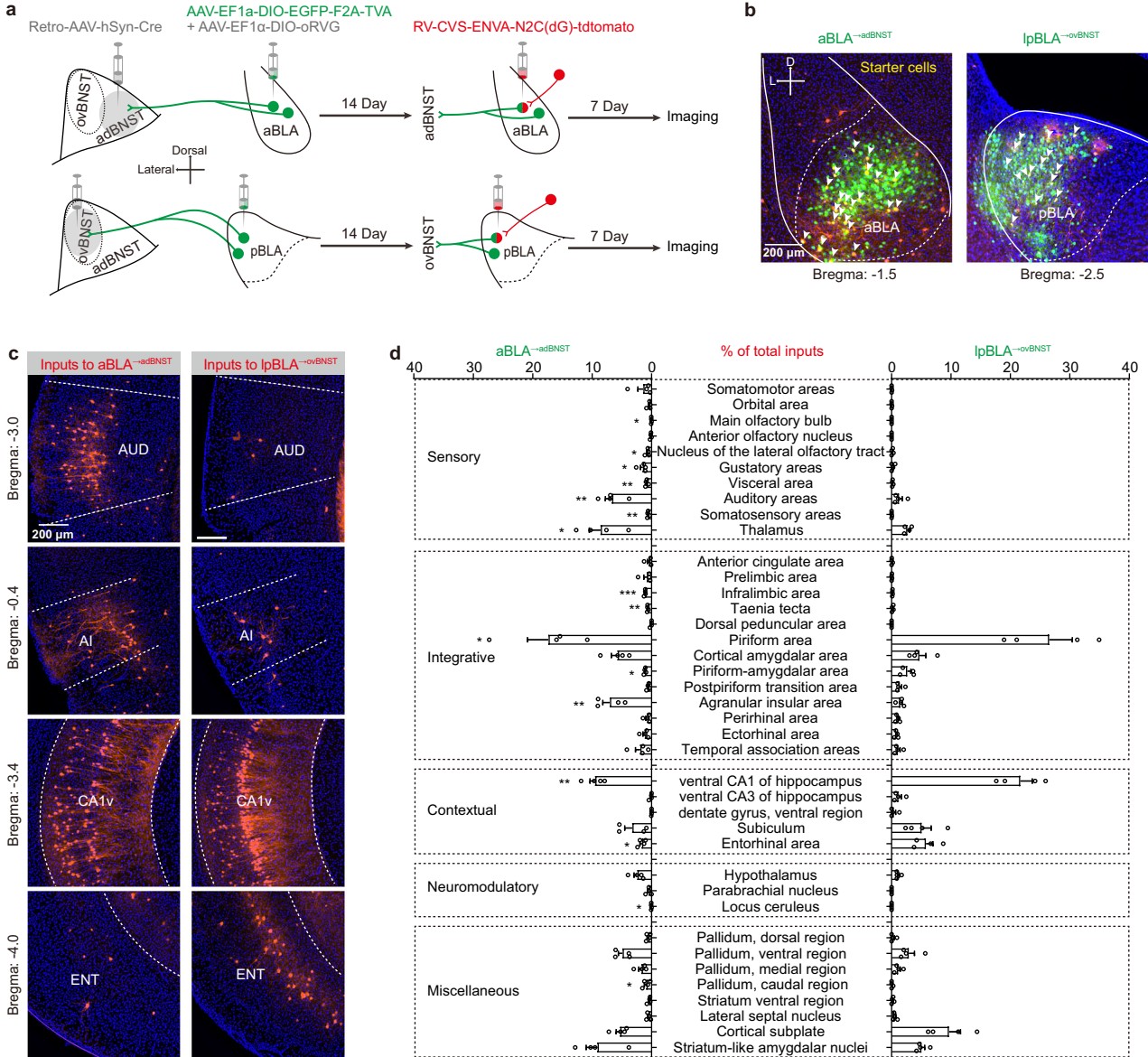

**Fig. 9 | Whole-brain mapping of the monosynaptic inputs to aBLA→adBNST and lpBLA→ovBNST PNs. a** Experimental procedures for whole-brain tracing of the monosynaptic inputs to aBLA→adBNST and lpBLA→ovBNST PNs using pseudotyped rabies. **b** Representative images showing the aBLA→adBNST (green, left) and lpBLA→ovBNST (green, right) PNs. The arrowed showing the start cells expressing RVG, TVA-EYFP and ENVA-N2C(dG)-tdtomato. This experiment was repeated 4 times with similar results. **c** Representative images showing the retrogradely labeled neurons in brain regions which exhibit distinct projections to aBLA→adBNST and lpBLA→ovBNST PNs. AI agranular insular area, AUD auditory areas, CA1v field CA1 ventral region, ENT entorhinal area. **d** Percentages of the retrogradely labeled cells in different upstream regions in ipsilateral sides of the injected sites. $n = 4$ mice. *$p < 0.05$, **$p < 0.01$, ***$p < 0.001$, multiple $t$-test. Data shown as means ± s.e.m. Source data including statistics are provided as a Source Date file.

CeC/L[22]. In conjunction with the findings from the present study, we are tempted to speculate that the BLA establishes segregated pathways to bidirectionally regulate the activity of certain target regions, thereby ensuring that their activity remains within an appropriate range as well as the associated behavioral and emotional outcomes. Nonetheless, the precise mechanisms underlying this regulation represent a crucial open question that warrants further investigation.

An increasing understanding of anxiety-engaging circuits has led to identification of numerous pathways, each playing variable roles in anxiety regulation[4,5,8,9]. However, a key question remains: how do these functionally divergent pathways interact to produce appropriate emotional and behavioral responses in the face of a constantly changing environment[1,3]? Our findings shed light on this issue by demonstrating extensive interactions between the two functionally opposing BLA→dBNST circuits, both at the start and target regions. These

extensive interactions may enable appropriate expression of anxiety in response to the ongoing demands of the environment.

In BLA, we found that activation of one subgroup of BLA→dBNST neurons, specifically the aBLA→adBNST or lpBLA→ovBNST PNs, significantly suppressed the activity of the other via strong reciprocal inhibition (Fig. 8). This mutual suppression prominently relies on the recruitment of local inhibitory network. First, activation of either PN subset resulted in a notable increase in the activity of local BLA interneurons. Second, blocking the GABAergic transmission abolished the reciprocal suppression between the two populations. Thus, the activity of each BLA→dBNST PN population depends not only on the strength of the external inputs they receive[58], but also on the lateral inhibition caused by the other subpopulation. This suppression effect establishes a lateral inhibitory motif that allow BLA to efficiently process anxiety-relevant information in the environment (Fig. 10). When external

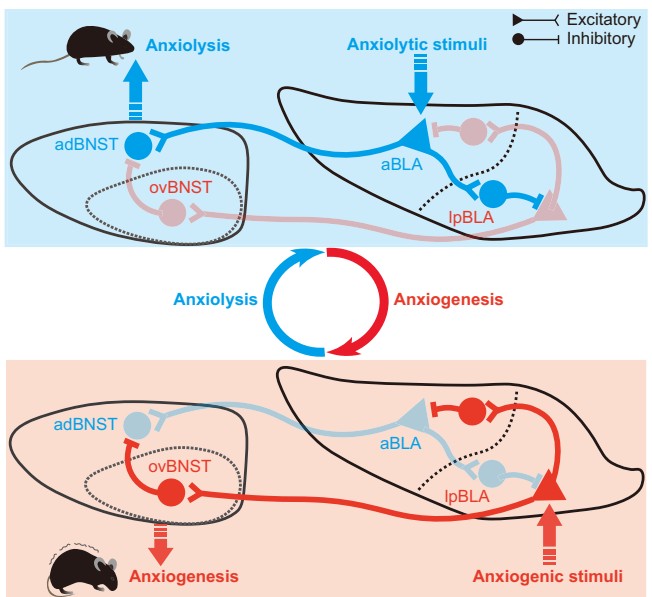

**Fig. 10 | A proposed model for the coordination between the two functionally opposing BLA→dBNST circuits.** In this model, we propose that when the anxiolytic information is transmitted to the BLA, it triggers the activation of aBLA^→adBNST PNs. Apart from eliciting anxiolytic behaviors by directly stimulating the adBNST neurons, aBLA^→adBNST PN activation also enhance the activity of the inhibitory interneurons in the lpBLA. The enhanced activation of these interneurons not only dampens the activity of anxiogenic lpBLA^→ovBNST PNs, but also relieves the inhibition imposed by the aBLA interneurons onto aBLA^→adBNST PNs, thereby further augmenting the activity of the latter. On the other hand, when the anxiogenic information is passed onto BLA, it activates lpBLA^→ovBNST PNs and elicits anxiogenic behaviors through engaging the inhibitory ovBNST→adBNST microcircuit. In addition, it also activates the interneurons in aBLA, which further suppress the activity of anxiolytic aBLA^→adBNST PNs with subsequent removal of their indirect inhibition onto lpBLA^→ovBNST PNs via disengaging the local interneurons.

information reaches BLA, it can directly activate one of the two BLA^→dBNST populations depending on the input regions carrying the information (Fig. 9). In addition, it can indirectly disinhibit this population by altering the balance of the cross inhibition between the two populations. These direct excitation and indirect disinhibition are expected to amplify the signal-to-noise ratio in the "winning" PN population, implementing a "winner-takes-all" mechanism during the processing of external information within the BLA. This mechanism may at least partially account for the consistent findings in our study (Fig. 7) and previous research, where certain BLA PN populations are selectively recruited or inhibited by cues charged with distinct valence[55,59–61]. The "winner-takes-all" strategy, alongside the extensive reciprocal inhibition among topographically and functionally divergent BLA PNs[27,30,32,59–62], may allow for the integration of information from various sources with varying emotional valences within the BLA[9,12,13].

In addition to the reciprocal inhibition between the two BLA→dBNST pathways in the start BLA region, they also communicate within the target dBNST region through strong ovBNST-to-adBNST feedforward inhibition (Fig. 3). The adBNST cells receive not only excitatory inputs from aBLA, but also robust feedforward inhibition following activation of lpBLA→ovBNST circuit. Moreover, multiple inputs that either reduce anxiety, such as mPFC, vHPC, or promote anxiety, such as CeA, lateral parabrachial nucleus were found to converge onto dBNST[63–66]. This region may thus emerge as a crucial hub for the integration and processing of both anxiolytic and anxiogenic information in the brain. Notably, unlike the symmetrical reciprocal inhibition between the two BLA→dBNST pathways in the start BLA region (Fig. 8), their interaction in the target dBNST seems to be asymmetrical, with the ovBNST-to-adBNST projection being far denser

than the adBNST-to-ovBNST projection[25,39,67]. Therefore, the robust inhibition from anxiogenic ovBNST neurons to their anxiolytic adBNST neighbors implies that the brain prioritizes the processing of unsafe and anxiogenic information, which may aid the survival of individuals in face of the constantly changing environment.

While the current findings have demonstrated that two functionally opposing BLA-to-dBNST circuits interact at multiple sites to fine tune anxiety-like behaviors, further investigations are needed to address some open important issues. For instance, the dBNST contains diverse neuronal subpopulations with distinct genetic, molecular, and anatomical characteristics[11,18]. The connectivity between the BLA subregions and these subpopulations, as well as the functional roles of these specific subcircuits, remain unknown. Additionally, besides their involvement in anxiety modulation, both the BLA and dBNST are implicated in regulating various emotions such as fear and reward, as well as essential behaviors for survival such as feeding and socializing[11,12,14,17,18,48]. It is intriguing to explore how the BLA and dBNST integrate diverse external cues and internal demands through their local and long-range interactions, ultimately leading to the selection of appropriate responses. Furthermore, both the BLA and dBNST exhibit sexual dimorphism in multiple aspects[68,69]. Therefore, it is imperative to investigate the connections and functions of these subcircuits between BLA and BNST in females.

## Methods

All experimental procedures were conducted in accordance with the guidelines of the National Institutes of Health and approved by the Institutional Animal Care and Use Committee of Nanchang University.

### Animals

Male C57BL/6 J mice (5–10 weeks) were used for all experiments. The mice were initially purchased from Model Animal Research Center of Nanjing University and bred in animal facility of Nanchang University. Mice were housed at 18–23 °C with 40–60% humidity in a 12-h light-dark cycle (light on: 6:00 a.m.–6:00 p.m.) with ad libitum access to food and water.

### Viruses

AAV-vGluT1-mCherry ($4.12 \times 10^{12}$ vg/ml, PT-5816), AAV-vGluT1-EYFP ($3.65 \times 10^{12}$ vg/ml, PT-5815), AAV-vGluT1-ChR2 (H134R)-EYFP ($5.83 \times 10^{12}$ vg/ml, PT-5813), AAV-vGluT1-GCaMP6s ($3.17 \times 10^{12}$ vg/ml, PT-5814), AAV-vGluT1-Cre ($2.69 \times 10^{12}$ vg/ml, PT-2950), retro-AAV-EF1a-mCherry ($5.43 \times 10^{12}$ vg/ml, PT-1940), retro-AAV-EF1a-Cre ($5.71 \times 10^{12}$ vg/ml, PT-0888), retro-AAV-DIO-ChR2-EYFP ($5.89 \times 10^{12}$ vg/ml, PT-0001), AAV-hSyn-GCaMP6s ($3.25 \times 10^{12}$ vg/ml, PT-0145), AAV-CaMKIIa-DIO-hM3D(Gq)-mCherry ($4.58 \times 10^{12}$ vg/ml, PT-1144), AAV-hSyn-NES-jRGECO1a ($3.58 \times 10^{12}$ vg/ml, PT-1593), AAV-vGluT1-NpHR-EYFP ($3.26 \times 10^{12}$ vg/ml, PT-9820), AAV-EF1a-DIO-EGFP-T2A-TVA ($5.97 \times 10^{12}$ vg/ml, PT-0062), AAV-EF1α-DIO-oRVG ($5.08 \times 10^{12}$ vg/ml, PT-0023) and RV-CVS-ENVA-N2C(dG)-tdtomato ($2.0 \times 10^{8}$ IFU/ml, RO5002). All viruses were purchased from BrainVTA Co., Ltd (China).

### Surgical procedures

Five-to-six-week-old mice were used for stereotaxic injections. Briefly, mice were anaesthetized with 2% pentobarbital sodium and placed in the stereotaxic frame (RWD, Shenzhen, China). Injection was performed using glass micropipettes with their tip diameters of ~10–20 μm (pulled with the Narishige PC-10 puller, Japan) mounted on 10-μL Hamilton Microlitre syringe (Hamilton Co., Reno, NV, USA). The AAV and Retrobeads were delivered at a rate of 10–20 nL/min using a stereotactic injector (QSI, Stoelting, Wood Dale, IL, USA), after which the pipettes were left in place for a further 5 min to allow diffusion.

To anterogradely label the axonal terminals of aBLA and pBLA PNs, AAV-vGluT1-EYFP (100 nL) and AAV-vGluT1-mCherry (100 nL) were

bilaterally infused into the aBLA (AP −1.5 mm, ML 3.15 mm, DV −5.0 mm) and pBLA (AP −2.5 mm, ML 3.25 mm, DV −5.1 mm), respectively.

To anterogradely label the axonal terminals of lpBLA and mpBLA PNs, AAV-vGluT1-EYFP (30 nL) was infused into either lpBLA (AP −2.5 mm, ML 3.40 mm, DV −5.1 mm) or mpBLA (AP −2.5 mm, ML 3.10 mm, DV −5.15 mm).

To retrogradely label the BLA PNs projecting to adBNST and ovBNST, retro-AAV-EF1a-mcherry (15 nL) was bilaterally infused into either the adBNST (AP + 0.35 mm, ML 0.8 mm, DV −4.15 mm) or ovBNST (AP + 0.15 mm, ML 1.0 mm, DV −4.0 mm). Alternatively, we also injected red and green Retrobeads (15 nL; R180, G180, LumaFluor) into the ovBNST and adBNST.

For in vitro optogenetic manipulations of the aBLA or pBLA inputs in the dBNST, AAV-vGluT1-ChR2-EYFP (100 nL) was bilaterally injected into either the aBLA or pBLA.

To investigate the interaction between aBLA$^{\rightarrow dBNST}$ and lpBLA$^{\rightarrow dBNST}$ neurons, retro-AAV-EF1α-mCherry and retro-AAV-DIO-ChR2-EYFP were mixed (1:1, 50 nL) and infused into dBNST (AP + 0.3 mm, ML 0.9 mm, DV −4.1 mm), then AAV-vGluT1-Cre was infused to either aBLA (100 nL) or lpBLA (30 nL) to selectively express ChR2 in aBLA$^{\rightarrow dBNST}$ or lpBLA$^{\rightarrow dBNST}$ neurons.

For in vivo optogenetic activation of the aBLA or pBLA inputs in the dBNST, AAV-vGluT1-hChR2-EYFP or AAV-vGluT1-EYFP (100 nL) was unilaterally injected into the aBLA or pBLA. 18 days later, an optical cannula (200-μm diameter, NA 0.22, Inper, Hangzhou, China) was ipsilaterally implanted over the dBNST (AP + 0.2 mm, ML 0.9 mm, DV −3.7 mm) to allow delivery of the blue light pulse to the axonal terminals of BLA PNs in the dBNST. To test the possible contribution from collateral projections, a guide cannula (26 G, RWD, Shenzhen, China) was implanted to the dBNST for drug infusion. About 30 min post drug infusion, the shell-removed optical fiber was inserted into the guide cannula and placed right above the dBNST for optogenetic manipulation. For in vivo optogenetic inhibition of the aBLA or pBLA inputs in the dBNST, AAV-vGluT1-NpHR-EYFP or AAV-vGluT1-EYFP (100 nL) was bilaterally injected into the aBLA or pBLA. 18 days later, two optical cannulas were bilaterally implanted over the dBNST to allow delivery of the yellow light to the axonal terminals of BLA PNs in the dBNST. To explore the specific function of mpBLA and lpBLA inputs in dBNST, AAV-vGluT1-hChR2-EYFP or AAV-vGluT1-EYFP (30 nL) were unilaterally injected into either mpBLA or lpBLA, followed by ipsilateral implantation of an optical cannula over the adBNST (AP + 0.35 mm, ML 0.8 mm, DV −3.8 mm) or ovBNST (AP + 0.15 mm, ML 1.0 mm, DV −3.7 mm), respectively.

To record the calcium activity of aBLA and lpBLA axonal terminals in dBNST in vivo, AAV-vGluT1-GCaMP6s was unilaterally infused into aBLA (100 nL) or lpBLA (30 nL). 18 days later, an optical cannula was ipsilaterally implanted over the adBNST or ovBNST to record the calcium activity of aBLA or lpBLA axon terminals, respectively.

To detect the effects of chemogenetic activation of aBLA$^{\rightarrow adBNST}$ and lpBLA$^{\rightarrow ovBNST}$ PNs on the activity of adBNST neurons in vivo, AAV-hSyn-GCaMP6s (15 nL) was unilaterally infused into the adBNST, and Retro-AAV-EF1a-Cre (30 nL) was ipsilaterally infused into dBNST to infect the input neurons in BLA. Then, AAV-CaMKIIa-DIO-hM3D(Gq)-mCherry (100 nL) was infused to either aBLA or pBLA to allow selective expression of hM3Dq in dBNST-projecting neurons in the two subregions. 18 days later, an optical cannula was ipsilaterally implanted over the adBNST for recording of the calcium activity in adBNST neurons. (AP + 0.35 mm, ML 0.6 mm, DV −4.0 mm; with a 20° angle to the vertical line).

To simultaneously detect the effects of photoactivation of aBLA$^{\rightarrow adBNST}$ and lpBLA$^{\rightarrow ovBNST}$ PNs on the activity of adBNST neurons and anxiety-like behaviors, AAV-hSyn-NES-jRGECO1a (15 nL) was unilaterally infused into the adBNST, and Retro-AAV-EF1a-Cre (30 nL) was ipsilaterally infused into dBNST to infect the input neurons in BLA. Then, AAV-vGluT1-ChR2-EYFP was infused to either aBLA (100 nL) or

lpBLA (30 nL) to allow selective expression of ChR2 in dBNST-projecting neurons in the two subregions. 18 days later, one optical cannula was implanted over aBLA or pBLA, the other optical cannula was ipsilaterally implanted over the adBNST.

To trace the monosynaptic inputs to aBLA$^{\rightarrow adBNST}$ and lpBLA$^{\rightarrow ovBNST}$ PNs, retro-AAV-EF1a-Cre (15 nL) was infused to the adBNST or ovBNST. Meanwhile, the mixed AAV vectors containing AAV-EF1a-DIO-EGFP-T2A-TVA and AAV-EF1α-DIO-oRVG (1:1, 100 nl) were injected into the aBLA or lpBLA (100 nL) correspondingly. 2 weeks later, RV-CVS-ENVA-N2C(dG)-tdtomato (100 nL) was injected into the same sites of aBLA or lpBLA. One week later, the animals were sacrificed for imaging.

## Optogenetic slice electrophysiology

### Slice preparation.
Mice were anesthetized with ether and decapitated, and the brains were rapidly removed and chilled in ice-cold, oxygenated (95% $O_2$ and 5% $CO_2$) cutting solution containing (in mmol/L) 80 NaCl, 3.5 KCl, 4.5 $MgSO_4$, 0.5 $CaCl_2$, 1.25 $NaH_2PO_4$, 25 $NaHCO_3$, 90 sucrose, and 10 glucose. Coronal slices containing the dBNST (250 μm) or BLA (300 μm) or horizontal slices containing aBLA and pBLA (300 μm) were prepared using the VT1000S Vibratome (Leica Microsystems, Wetzlar, Germany). The slices were placed in artificial cerebrospinal fluid (ACSF) containing (in mmol/L) 124 NaCl, 2.5 KCl, 2 $MgSO_4$, 2.5 $CaCl_2$, 1.25 $NaH_2PO_4$, 22 $NaHCO_3$, and 10 glucose and was bubbled with 95% $O_2$ and 5% $CO_2$ for 30 min at 34 °C, and then maintained at room temperature for at least 1 h before recordings.

### Whole-cell recording.
Whole-cell patch clamp recordings were performed by using an infrared differential interference contrast microscope (BX51WI, Olympus, Tokyo, Japan) equipped with two automatic manipulators (Sutter Instrument Co., Novato, CA) and a highly sensitive CCD camera (IR-1000E, DAGE-MTI, Michigan, IN, USA). The brain slices were transferred to the recording chamber and continuously perfused with oxygenated ACSF at a rate of ~2 ml/min. The temperature of ASCF was maintained at 29 ± 1 °C with an automatic temperature controller (TC-324B, Warner Instrument Co. Hamden, CT). Recording electrodes were made from filamented borosilicate glass capillary tubes (inner diameter, 0.84 μm) by using a horizontal pipette puller (P-97; Sutter Instrument Co., Novato, CA). In voltage-clamp, the pipettes with resistance ranged from 4 to 7 MΩ were filled with intracellular solution containing (in mM): 130 Cs-methanesulfonate, 5 NaCl, 1 MgCl2, 10 HEPES, 0.2 EGTA, 5 QX314, 2 ATP-Mg, and 0.1 GTP-Na (pH: 7.3–7.4; osmolarity: 290 mOsm). In current-clamp, the patch electrodes with resistance ranged from 4 to 7 MΩ were filled with pipette solution contained (in mmol/L) 130 K-gluconate, 5 KCl, 1 MgCl2, 10 HEPES, 0.2 EGTA, 2 ATP-Mg, 0.1 GTP-Na (pH: 7.3–7.4; osmolarity: 290 mOsm). A junction potential of ~12 mV was uncorrected. Data were sampled at 10 kHz filtered at 2 kHz using the patch-clamp amplifier (EPC 10 USB, HEKA Instrument, Germany) circuitry and collected with the PATCHMASTER software (version 2.53). Series resistance (Rs) was in a range of 10–20 MΩ and monitored throughout the experiments. If Rs changed >20% during recording, the data were excluded. Offline data analysis was performed using Origin 8.5 (Microcal software, Northampton, MA, USA).

### Optogenetics in brain slice.
To optogenetically activate the ChR2-expressing BLA PNs or their axonal terminals in dBNST, the blue light pulses were generated by using a light-emitting diode (LED) with 470 nm peak wavelength (M470L3, Thorlabs Inc., Newton, NJ, USA) connected to a Master-8 Pulse Stimulator through the LED driver (LEDD1B, Thorlabs Inc.). The light was delivered to the brain slices through a 40× water-immersion objective lens (40×/NA0.8, LUM-PlanFL, Olympus) and its intensity was measured with Optical Power Meter (PM100D power meter, Thorlabs Inc.).

To investigate the impact of aBLA and pBLA inputs activation on the firing of their target cells in dBNST, we delivered a low-intensity

 

depolarizing current (-15 pA) to the recorded cells in adBNST or ovBNST to evoke their firing at ~3 Hz. The ChR2-expressing aBLA or pBLA terminals were then illuminated with a train of light pulses (10 Hz, 1 ms, 20 μW/mm²) for 2 s and their effects on the neuronal firings were tested. The membrane potentials were held at −70 mV before current injection.

Similar procedures were used to measure the interaction between aBLA$^{→dBNST}$ and lpBLA$^{→dBNST}$ PNs. To detect the effect of aBLA$^{→adBNST}$ neurons on the activity of lpBLA$^{→dBNST}$ PNs or the interneurons in lpBLA, we performed current-clamp recordings on lpBLA PNs or interneurons and delivered a depolarizing current to the recorded cells (-100 pA for PNs and -15 pA for interneurons) to evoke their spiking at ~3 Hz. The aBLA$^{→adBNST}$ PNs were then optogenetically activated and their effects on the recorded cells were measured. With reference to the above procedures, the effect of lpBLA$^{→ovBNST}$ PNs on the activity of aBLA$^{→adBNST}$ PNs and interneurons in aBLA was also investigated.

To construct the input–output curves of synaptic responses of dBNST neurons against the intensity of light pulses delivered to aBLA or pBLA inputs, we recorded the oEPSCs and oIPSCs by holding the recorded cells at −70 and 0 mV, respectively. Light pulses of increasing intensity (5, 10, 20, 50, and 100 μW/mm², 1 ms) were delivered at an interval of 10 s. The latencies of oEPSCs and oIPSCs were calculated as time interval between the start of stimulation and the onset of currents.

**Optogenetics with focalized light stimuli in brain slice.** To selectively activate the pBLA axonal terminals in either adBNST or ovBNST, we delivered a focalized light spot (~120 μm in diameter) generated by a Polygon DMD device (Mightex) to either of the two dBNST subregions at a random order. Their effects were then examined on the firing or synaptic responses of the adBNST cells.

### In vivo optogenetics and behavioral assays

All mice underwent a 3−5-minute daily handling by an experimenter for three consecutive days before behavioral testing to minimize any stress introduced by the experimenter. On the test day, the optical fiber was connected to a blue laser (470 nm, 10 Hz pulses, 5 ms duration, Newdoon Inc.) for stimulation. The LED power measured at the tip of the optical fiber was 3−5 mW. Five minutes were allowed for recovery in the home cage from handling associated with connecting the optical patch cable before the session.

**The elevated plus maze test (EPM).** The maze apparatus consisted of two opposing open (35 cm × 6 cm) and two enclosed arms (35 cm × 6 cm) extending from a central platform (6 cm × 6 cm), and raised 74 cm above the floor. During the test, mice were placed in the center platform, facing an open arm, followed by a 9-min monitoring of their behavior. The 9-min session was divided into three 3-min epochs, the light-off epoch, the light-on epoch and the light-off epoch (off-on-off epochs). A video-tracking system (ANY-maze, Stoelting) was used to automatically track and analyze their entries into the open arms and the time they spent in the open arms. The apparatus was cleaned with 30% ethanol after each trial.

**The open field test (OFT).** The open field chamber was made of transparent plastic (50 cm × 50 cm). Individual mice were placed in the center of the chamber and their behavior was monitored for 9 min (off-on-off epochs, 3 min for each epoch) with an overhead video-tracking system.

### Fiber photometry

The optical fiber was connected to the fiber photometry system (488 nm, Thinker Tech Nanjing Biotech Limited Co., Ltd). Calcium fluorescence was captured using a photomultiplier tube and subsequently converted into electrical signals at a rate of 100 Hz. Animal behavior was monitored using an overhead video-tracking system. In order to synchronize the analysis of calcium signals and behaviors, their activities were simultaneously recorded using screen capture software. (EVCapture, ieway information technology Co. LTD, China).

To investigate the effect of activating aBLA and pBLA inputs on adBNST neuronal activity in vivo, each mouse was individually housed in its home cage for 5 days before recording to minimize stress. During the experiment, a 20-minute baseline measurement of calcium signal was obtained prior to intraperitoneal injection of either the vehicle or CNO (3 mg/kg), followed by continuous recording of calcium signal for 60 min after the treatment. To reduce stress associated with drug administration, mice were lightly anesthetized with isoflurane during the injection. To minimize GCaMP bleaching, the laser intensity was set to a low level of 20 μW at the tip of the optical fiber. Moreover, in order to mitigate the impact of bleaching, the calcium signal for each mouse was calibrated using MATLAB-based software associated with the fiber photometry system. To compare the effects of vehicle and CNO treatment within each mouse, both treatments were sequentially administered in a randomized order with a 4-day interval. The order of vehicle and CNO treatments was counterbalanced within each group. Data were analyzed using MATLAB-based software affiliated with the fiber photometry system. Calcium signal values were presented as dF/F0 (calculated as $(F - F0)/F0 × 100\%$), where F0 represents the average calcium signal during the baseline period of 20 min before intraperitoneal injection. Therefore, the average dF/F0 value during the baseline period was zero. The average calcium signal during the 60-minute period following treatment was compared between the two treatment conditions within each mouse. The calcium events with more than 2.5MAD (mean absolute deviation) of the baseline session were automatically extracted with MATLAB-based software. The frequency and the average peak amplitude of the events following treatment were normalized to these before treatment.

To detect the effect of aBLA$^{→adBNST}$ and lpBLA$^{→ovBNST}$ PNs activation on the adBNST neural activity and anxiety-like behaviors in both OFT and EPM, the calcium signal of adBNST neurons was recorded during the behavioral assay. $Ca^{2+}$ fluorescence values were indicated by z-score. Here, the average $Ca^{2+}$ signal in a 10-s-long period before light-on was set as baseline. Z-score values are presented as heatmaps. The average $Ca^{2+}$ signal during a 180-second period following the onset of light stimulation was compared to that during a 30-second period prior to light stimulation.

To detect the real-time changes in the activity of the aBLA and lpBLA fibers in adBNST and ovBNST respectively, mice were placed in the center platform of EPM, facing an open arm. Subsequently, their behavior and calcium signal were recorded for a continuum of 10 min. $Ca^{2+}$ fluorescence values were indicated by z-score. Here, the average $Ca^{2+}$ signal in a 5-s-long period before entering to open or closed arms was set as baseline. Time 0 s was set when all the four paws of the tested mouse entered the open or closed arm. Z-score values are presented as heatmaps and also as average plots with a shaded area indicating the s.e.m. The $Ca^{2+}$ signal in a 10-s-long period after entry was compared to the baseline.

### Histology and microscopy

Mice were anesthetized with 2% pentobarbital sodium and then transcardially perfused with ice-cold 0.1 M phosphate-buffered saline (PBS) followed by 4% paraformaldehyde (PFA). Brains were post-fixed overnight at 4 °C in 4% PFA and then coronal slices (50-μm thick) containing the dBNST and BLA were sectioned with VT1000S Vibratome (Leica Microsystems, Wetzlar, Germany). Slices were then mounted onto the slides with fluoromount aqueous mounting medium containing DAPI (4′,6-diamidino-2-phenylindole, a DNA-specific fluorescent probe; Beyotime, P0131, China). Immunofluorescence images were taken by using a high-throughput microscope (Olympus VS120, Tokyo, Japan). Brain areas were determined by anatomical landmarks (such as anterior commissure, lateral ventricle,

hippocampus) and were based on the Mouse Brain Atlas in Stereotaxic Coordinates, second edition. Only the mice with appropriate virus expression and optical cannula position were used.

## c-Fos immunohistochemical staining

Mice expressing hM3Dq were administered with either vehicle or CNO (3 mg/kg). 90 min later, mice were anesthetized with 2% pentobarbital sodium and transcardially perfused with ice-cold 0.1 M PBS followed by 4% PFA. Coronal brain sections (50 μm) containing aBLA or pBLA with hM3Dq expression were prepared. Then, the sections were washed three times in PBS (3 × 5 min), and blocked in permeable buffer (0.1% Triton X-100 in PBS, PBST) containing 10% normal goat serum for 2 h at room temperature. Sections were incubated with the primary antibody against c-Fos (1:500; Cell signaling, 2250) overnight at 4 °C, followed by three washes in PBST and incubation with the fluorescent secondary antibody (donkey anti-rabbit Alexa Fluor 488; 1:1000, Invitrogen, A21206) at room temperature for 2 h. Sections were washed three times in PBST before being mounted onto the slides with Fluoromount Aqueous Mounting Medium containing DAPI. Confocal immunofluorescence images were taken by using a scanning laser microscope (Olympus FV1000, Tokyo, Japan).

## Statistics

Sample sizes were determined based on previous experience and related literature. For the physiological experiments, investigators were blinded to group allocation during data collection and analysis. The behavioral data were collected and analyzed using computer software in an unbiased manner, rendering blinding unnecessary. Statistical analyses were performed by Prism 8.0.2 (GraphPad). All data were presented as means ± s.e.m. One-way analysis of variance (ANOVA) or two-way ANOVA followed by Bonferroni's multiple comparisons test performed for more than two groups, while two tailed independent samples $t$-test or paired $t$-test were used to compare the differences between two groups. Two tailed one sample $t$-test was used to compare the differences of one group with a theoretical mean. Statistical analysis used, n, and parameters for determining significance are described in the figure legends. The homoscedasticity and normality of the distributions were analyzed with Bartlett's and Kolmogorov–Smirnov tests, respectively. The threshold for statistical significance was set at $p < 0.05$.

## Reporting summary

Further information on research design is available in the Nature Portfolio Reporting Summary linked to this article.

## Data availability

Source data are provided with this paper.

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

## Acknowledgements

This work was supported by grants from National Natural Science Foundation of China (Grant Nos. 82125010 and 81930032 to P.B., 82160270 to R.-W.H.,) and National Key R&D Program of China (2021ZD0202704 to B.-X.P.).

## Author contributions

R.-W.H. and B.-X.P. conceived the study. R.-W.H., Z.-Y.Z., C.J. and Z.-Y.H. performed the experiments. R.-W.H., Z.-Y.Z. and C.J. analyzed the data. R.-W.H. and B.-X.P. wrote the manuscript. All authors read and approved the final manuscript.

## Competing interests

The authors declare no competing interests.
