## [Peer Review File · Nature Communications]

Synergism between two BLA-to-BNST pathways for appropriate expression of anxiety-like behaviors in male miceREVIEWER COMMENTS

Reviewer #1 (Remarks to the Author):

In this manuscript, Han et al. conduct an extensive study on the interplay between two distinct sub-circuits linking the Basolateral Amygdala (BLA) and the Bed Nucleus of the Stria Terminalis (BNST). They demonstrate that these pathways have opposing roles in the regulation of anxiety. Specifically, they show that the anterior BLA (aBLA) preferentially innervates the anterodorsal BNST (adBNST), while the posterior BLA (pBLA) mainly targets the oval BNST (ovBNST). Activation of the aBLA to adBNST pathway induces anxiety, whereas activation of the pBLA to ovBNST pathway alleviates it. Furthermore, they reveal that projection neurons from the aBLA and pBLA to the BNST inhibit each other via local interneurons. These findings significantly enhance our understanding of the interaction between BLA and BNST in anxiety regulation.

While the study is robustly supported by well-executed experiments and compelling behavioral effects, it lacks direct evidence to show the impact of each BLA input on neuronal activity in the BNST. To address this gap, the authors could combine recordings of neuronal activity in the BNST with targeted manipulation of each BLA input during Open Field Test (OFT) and Elevated Plus Maze (EPM) experiments. This approach would also allow for the testing of various hypotheses presented in the discussion section in the manuscript and could elucidate the potential role of pBLA-driven feedforward inhibition from the ovBNST to the adBNST in regulating neuronal activity in anxiety-inducing conditions.

Another major concern is the manuscript's reliance on gain-of-function experiments. Incorporating loss-of-function experiments would provide a more comprehensive view of the endogenous roles of each pathway in anxiety regulation.

Additionally, all figures in the manuscript are missing labels indicating their location along the Anteroposterior (AP) axis. Including this information would aid readers in identifying the anatomical context of each representative brain image.

Reviewer #2 (Remarks to the Author):

Summary:

The research was conducted with solely male mice, and the authors did not provide justifications.

First, the authors identified anatomically distinct projections, aBLA -> adBNST) and pBLA -> ovBNST) with moderate projection to adBNST. They further segregate the latter into mpBLA->adBNST and lpBLA->adBNST. Each projection also provides a distinct synaptic response and effects on dBNST microcircuit activities. Interestingly, both brain sites, BLA and BNST, have a reciprocal mutual inhibition system between each subnucleus. Second, the authors tried to correlate the identified circuit properties to anxiety behaviors, the open field and elevated plus maze by two means. One is activating each projection by optogenetics methods, and another is measuring BLA terminal activities in BNST by photometry. Lastly, they examined the upstream brain structures from each subnucleus of BLA.

The story reminds the reviewer of a series of studies from the Tonegawa Lab showing genetically identified two types of BLA neurons distributed opposingly along the anteroposterior axis and exhibiting opposing behavioral functions through projection to CeA (1, 2). The authors discovered similar bidirectional control by two subpopulations of BLA neurons projecting to BNST for anxiety behavior. Unlike in Tonegawa's studies, the authors relied on anatomical segregation into subnuclei instead of using genetic markers and added the synaptic/circuit analyses through slice physiology.

The significance of this paper is in demonstrating the relevance of the BLA-BNST circuitry during the actual anxiety behaviors, which was done by in vivo optometry recording during solely EPM, not in the open field and is supported by testing several artificial optogenetic stimulations of each projection on the behaviors.

Below, the reviewer lists the issues that challenge the abilities of the presented data to support the authors' conclusions.

Major:

1. Clear explanations are required regarding data handling in Fig6.

In EPM, the transition between arms, particularly from the closed to the open arm, is slow and varies among animals and trials.

1) The authors are requested to explain why Fig6d and l show clear transitions with small error bars at the time 0s and to clarify their behavioral definition of animals in closed and open arms.

2) Some animals show sharp changes in the averaged data. It is required to report the sample numbers of each transition for each animal in Fig6e, g, m and o.

3) Since data are collapsed among multiple animals, in particular, the virus expression of which varies, processing the results as z score is necessary.

2. Artificial optogenetic stimulation may activate collateral projections through backpropagation.

There are chances for the BLA neurons to have collateral projections to other brain regions. Please rule out the possibility of activating other brain regions through action potentials that backpropagate to the cell body to contribute to the observed behavior changes.

Minor:

Fig1d. Why do the normalized adBNST values have variations?

Fig2m, n, p and q. In Fig2n and q, the difference of averaged %dF/F for 60 min is small relative to the peak heights in the traces. The authors are requested to show the entire traces. The disclosure will also help to show the degree of GCaMP bleaching during 60 min.

Fig3b,c and i. Given the variability of virus expression among brain slices and animals, comparing samples based on the light power without internal control is impossible.

In addition, in Fig3i, the authors used focalized light stimulation with 120 um in diameter; however, the x-axis ranges are the same among Fig3b,c and i. Clarification is requested for scientific integrity.

Fig 7b and k. In Fig7b, it's hard to see yellow in aBLA, whereas in Fig7k, it's hard to see red in aBLA. The authors may provide magnified images that show the expected co-expression of colors.

1. Kim J, Zhang X, Muralidhar S, LeBlanc SA, Tonegawa S (2017): Basolateral to Central Amygdala Neural Circuits for Appetitive Behaviors. *Neuron*. 93:1464-1479 e1465.

2. Kim J, Pignatelli M, Xu S, Itohara S, Tonegawa S (2016): Antagonistic negative and positive neurons of the basolateral amygdala. *Nat Neurosci*. 19:1636-1646.

Reviewer #3 (Remarks to the Author):

This manuscript explores how different aspects of the BLA innervate the BNST and how that impacts behavior. The study is very well done and rigorous. The authors find that anterior BLA drives reductions in anxiety-like behavior, while the posterior drives increases. The authors then use a careful slice physiology approach to define how this occurs, building on many previous papers. This is a really outstanding body of work that will be of high interested to the field.

Responses to the reviewers' comments

We thank all the three reviewers for their highly valuable comments, which we have incorporated into the revised version to improve the overall quality of this manuscript.

Reviewer #1 (Remarks to the Author):

In this manuscript, Han et al. conduct an extensive study on the interplay between two distinct sub-circuits linking the Basolateral Amygdala (BLA) and the Bed Nucleus of the Stria Terminalis (BNST). They demonstrate that these pathways have opposing roles in the regulation of anxiety. Specifically, they show that the anterior BLA (aBLA) preferentially innervates the anterodorsal BNST (adBNST), while the posterior BLA (pBLA) mainly targets the oval BNST (ovBNST). Activation of the aBLA to adBNST pathway induces anxiety, whereas activation of the pBLA to ovBNST pathway alleviates it. Furthermore, they reveal that projection neurons from the aBLA and pBLA to the BNST inhibit each other via local interneurons. These findings significantly enhance our understanding of the interaction between BLA and BNST in anxiety regulation.

We thank the reviewer for the thorough evaluation of our manuscript and the positive comments.

While the study is robustly supported by well-executed experiments and compelling behavioral effects, it lacks direct evidence to show the impact of each BLA input on neuronal activity in the BNST. To address this gap, the authors could combine recordings of neuronal activity in the BNST with targeted manipulation of each BLA input during Open Field Test (OFT) and Elevated Plus Maze (EPM)

experiments. This approach would also allow for the testing of various hypotheses presented in the discussion section in the manuscript and could elucidate the potential role of pBLA-driven feedforward inhibition from the ovBNST to the adBNST in regulating neuronal activity in anxiety-inducing conditions.

To provide direct evidence to show the impact of each BLA input on neuronal activity in the adBNST, we performed new experiments in the revised manuscript. Following the reviewer's suggestion, we tested the changes of adBNST neuronal activity in response to the activation of either aBLA or pBLA input during OFT and EPM. The results (**Fig. 6 of the revised manuscript**) show that activation of aBLA \rightarrow adBNST PNs enhances the activity of adBNST neurons while reducing the mice's anxiety-like behavior in OFT and EPM. Conversely, activation of lpBLA \rightarrow ovBNST PNs inhibits the activity of adBNST cells with increases of mice's anxiety-like (**Page 13-14, Line 254-270**).

Another major concern is the manuscript's reliance on gain-of-function experiments. Incorporating loss-of-function experiments would provide a more comprehensive view of the endogenous roles of each pathway in anxiety regulation.

We appreciated the reviewer for this constructive comment. Following the reviewer's suggestion, we have performed new experiments to explore how optogenetically inhibiting each input would affect the mice's anxiety-like behaviors. We found that optogenetic inhibition of the aBLA \rightarrow adBNST circuit increases the anxiety-like behaviors of the mice in both OFT and EPM, while inhibition of the pBLA \rightarrow adBNST circuit reduces these behaviors (**Supplementary Fig. 7, Page 12, Line 220-223**).

Additionally, all figures in the manuscript are missing labels indicating their location along the Anteroposterior (AP) axis. Including this information would aid readers in identifying the anatomical context of each representative brain image.

We are sorry for missing the labels indicating their locations along the AP axis, which have been added to all figures in the manuscript.

Reviewer #2 (Remarks to the Author):

Summary:

We appreciate the reviewer for the thorough evaluation of our manuscript and all the insightful and constructive comments that helped strengthening the story (see below).

The research was conducted with solely male mice, and the authors did not provide justifications.

As pointed out by the reviewer, this research was solely conducted with male mice, which we have clarified in the title of this manuscript.

First, the authors identified anatomically distinct projections, aBLA) -> adBNST) and pBLA) -> ovBNST) with moderate projection to adBNST. They further segregate the latter into mpBLA->adBNST and lpBLA->adBNST. Each projection also provides a distinct synaptic response and effects on dBNST microcircuit activities. Interestingly, both brain sites, BLA and BNST, have a reciprocal mutual inhibition system between each subnucleus. Second, the authors tried to correlate the identified circuit properties to anxiety behaviors, the open field and elevated plus maze by two means. One is activating each projection by optogenetics methods, and

another is measuring BLA terminal activities in BNST by photometry. Lastly, they examined the upstream brain structures from each subnucleus of BLA.

The story reminds the reviewer of a series of studies from the Tonegawa Lab showing genetically identified two types of BLA neurons distributed opposingly along the anteroposterior axis and exhibiting opposing behavioral functions through projection to CeA (1, 2). The authors discovered similar bidirectional control by two subpopulations of BLA neurons projecting to BNST for anxiety behavior. Unlike in Tonegawa's studies, the authors relied on anatomical segregation into subnuclei instead of using genetic markers and added the synaptic/circuit analyses through slice physiology.

The significance of this paper is in demonstrating the relevance of the BLA-BNST circuitry during the actual anxiety behaviors, which was done by in vivo optometry recording during solely EPM, not in the open field and is supported by testing several artificial optogenetic stimulations of each projection on the behaviors.

Below, the reviewer lists the issues that challenge the abilities of the presented data to support the authors' conclusions.

Major:

1. Clear explanations are required regarding data handling in Fig6.

In EPM, the transition between arms, particularly from the closed to the open arm, is slow and varies among animals and trials.

1) The authors are requested to explain why Fig6d and I show clear

transitions with small error bars at the time 0s and to clarify their behavioral definition of animals in closed and open arms.

As stated by the reviewer, the error bars at the time 0s (also during 5s before the mice transitioned from the closed to the open arms) in **Fig 6d and i (the Fig 7d and i of the revised manuscript)** are relatively smaller than these during 0-10 s when mice explored in the open arms. The reason for the small error bars during -5 to 0s (including 0 s) may arise from the data processing in which we set the average z-score value of each trial during the period from -5 to 0s as the baseline (0). For instance (please see the Figure below), when we randomly selected a group of trials ($n = 26$) and set -5 to 0 s as the baseline, the error bars during -5 to 0 s is relatively smaller than these during 0-10s. Time 0 s was set when the front half of the mouse's body enters the open or closed arm, which has been clarified in the Methods (**Page 36, Line 720-721**).

2) Some animals show sharp changes in the averaged data. It is required to report the sample numbers of each transition for each animal in Fig6e, g, m and o.

As suggested by the reviewer, we have included the sample number of transitions for each animal in **Fig 6 e, g, m and o (Fig 7 of the revised manuscript, Source data)**. The sharp changes in some animals should contribute to the relatively large error bars during mice's exploration in the open arms.

3) Since data are collapsed among multiple animals, in particular, the virus expression of which varies, processing the results as z score is necessary.

Thanks for this constructive comment and we have processed the results as z-score.

2. Artificial optogenetic stimulation may activate collateral projections through backpropagation.

There are chances for the BLA neurons to have collateral projections to other brain regions. Please rule out the possibility of activating other brain regions through action potentials that backpropagate to the cell body to contribute to the observed behavior changes.

We agree with the reviewer that there are chances of activating the collateral projections during stimulation of BLA efferents. To test the possible involvement of the collateral projections to other brain regions in the anxiety-modulatory effects of BLA→dBNST pathways, we delivered CNQX and AP5, the glutamate receptor antagonists or the vehicle solutions to dBNST 30 min prior to photoactivation of aBLA or pBLA terminals (1, 2). We found that pretreatment with

CNQX and AP5 but not vehicle solution deprived the modulatory effects of both BLA→dBNST pathways in regulating the anxiety-like behaviors in both OFT and EPM (**Supplementary Fig 6, Page 11, Line 208-219,**). These findings strongly suggest that the direct aBLA and pBLA inputs to dBNST are required for the anxiety regulation of BLA→dBNST pathways, and the roles of their collateral projections, if there are some, is insufficient to mediate the anxiety-modulatory effects.

1. Felix-Ortiz, A.C. *et al.* BLA to vHPC inputs modulate anxiety-related behaviors. *Neuron* **79**, 658-664 (2013).
2. Pi, G. *et al.* Posterior basolateral amygdala to ventral hippocampal CA1 drives approach behaviour to exert an anxiolytic effect. *Nat Commun* **11**, 183 (2020).

Minor:

Fig1d. Why do the normalized adBNST values have variations?

The reason why the normalized adBNST (ovBNST) values show variations is because we set the average adBNST (ovBNST) value of all the 4 tested mice rather than the corresponding values of each mice as 1. The original data were shown in **Source data**.

Fig2m, n, p and q. In Fig2n and q, the difference of averaged %dF/F for 60 min is small relative to the peak heights in the traces. The authors are requested to show the entire traces. The disclosure will also help to show the degree of GCaMP bleaching during 60 min.

As pointed out by the reviewer, the difference of averaged %dF/F is small relative to the peak heights in the traces. Although the peak changes for each trace are 10-20%, the average values of these

traces are around 4 %. And the difference of these average values between vehicle or CNO application are much smaller (-1-1% for most animals).

As claimed by the reviewers, there is indeed some degree of GCaMP bleaching during long term recording. To minimize the impact of fluorescent bleaching, we used relatively low excitation light power during the experiment. The collected data were calibrated using MATLAB-based software associated with the fiber photometry system (**Page 33, Line 692-696**). We included the calibrated data in **Supplementary Fig 2**. Also, the original data for one trace were attached below for the reviewer's references.

Fig 3b,c and i. Given the variability of virus expression among brain slices and animals, comparing samples based on the light power without internal control is impossible.

We agree with the reviewer that without internal control, comparisons among the brain slices and animals with variable virus expression is impossible. For **Fig. 3b and c**, we have re-analyzed

the data by making comparisons of the amplitudes of oEPSCs and oIPSCs (recorded from the same cells in the identical slices) using repeated measures two-way ANOVA (**Page 8-9, Line 154-157**).

For the comparison in **Fig 3i**, the two forms of oIPSCs were recorded from the same cells in the same slices only when the light spots were placed in adBNST and ovBNST respectively, Comparison of their strength may help understanding the relative contribution of local interneuronal network in adBNST versus the distal network in ovBNST to the oIPSCs in adBNST cells.

In addition, in Fig3i, the authors used focalized light stimulation with 120 um in diameter; however, the x-axis ranges are the same among Fig3b,c and i. Clarification is requested for scientific integrity.

We appreciate the reviewers for pointing out this error. Actually we set the same light intensity (but not the light power) for the recording conditions in **Fig 3b, c and i**. We are sorry for the misleading caused and make correction in the revision version.

Fig 7b and k. In Fig7b, it's hard to see yellow in aBLA, whereas in Fig7k, it's hard to see red in aBLA. The authors may provide magnified images that show the expected co-expression of colors.

We have followed the reviewer's suggestion and provide magnified images that show the expected co-expression of colors.

Reviewer #3 (Remarks to the Author):

This manuscript explores how different aspects of the BLA innervate the BNST and how that impacts behavior. The study is very well done and rigorous. The authors find that anterior BLA drives reductions in anxiety-like behavior, while the posterior drives

increases. The authors then use a careful slice physiology approach to define how this occurs, building on many previous papers. This is a really outstanding body of work that will be of high interest to the field.

We appreciate the reviewer for his/her thorough evaluation of our manuscript and the positive comments.

REVIEWER COMMENTS

Reviewer #1 (Remarks to the Author):

The authors have addressed all my concerns with new experiments and analysis. Now the manuscript is ready to publish. This is a very nice work. Congratulations.

Reviewer #2 (Remarks to the Author):

The reviewer appreciated the author's responses to the comments.

1. Definition of entry to the open/closed arm

The author: "Time 0 s was set when the front half of the mouse's body enters the open or closed arm, which has been clarified in the Methods (Page 36, Line 720-721)."

The reviewer: The well-accepted definition of an arm entry is "all four paws must enter the arm." The author is requested to 1) report the results by recomputing using the strict definition, 2) add results on the distance to the center of mass of animals from a landmark of the maze vs. time from all subjects together with the average. In addition, when animals are in the half-body at the boundary, particularly at the entry of the open arm, they hesitate to go to the open arm. The author is requested to 3) report data in both cases: they go to the open arm and return to the closed arm.

2. Ruling out the possibility of contribution from collateral projections

The reviewer: CNQX and AP5 infusion into the BNST is anxiolytic (Kim et al., 2013, see below). The author provided new results, supplementary Fig6, which indicate no anxiolytic effect with the treatment in the open field and elevated plus maze. The reviewer understands implanting the optical cannula (200um diameter) and the cannula for infusion (26G RWD, 457um diameter) is challenging. The associated histology does not match the expected tissue damage from optical and infusion cannulae (657um).

The author is requested to explain the two discrepancies above and describe the details of the implants.

Kim, Sung-Yon, Avishek Adhikari, Soo Yeun Lee, James H. Marshel, Christina K. Kim, Caitlin S. Mallory, Maisie Lo, et al. "Diverging Neural Pathways Assemble a Behavioural State from Separable Features in Anxiety." *Nature* 496, no. 7444 (April 2013): 219–23. <https://doi.org/10.1038/nature12018>.

3. Data handling of optometry results in Figure 2

The reviewer restates the concern "Fig2m, n, p and q. In Fig2n and q, the difference of averaged %dF/F for 60 min is small relative to the peak heights in the traces". The author's raw photometry data shows an unstable baseline as regular, originating from photobleaching and locomotion. Calibration of the base with curve/linear fitting inevitably included a substantial error when comparing the values from averaged signals, particularly when the author sought to detect small differences (0.3 in Fig2n, 0.7 in Fig2q). The author is requested to report the average amplitudes and frequencies of the transients to avoid such errors.

Reviewer #3 (Remarks to the Author):

the authors have adequately addressed all of the comments from the previous review.

Responses to the reviewers' comments

We thank all the three reviewers for their highly valuable comments, which we have incorporated into the revised version to improve the overall quality of this manuscript.

REVIEWER COMMENTS

Reviewer #1 (Remarks to the Author):

The authors have addressed all my concerns with new experiments and analysis. Now the manuscript is ready to publish. This is a very nice work. Congratulations.

We thank the reviewer for the positive evaluation of our manuscript.

Reviewer #2 (Remarks to the Author):

The reviewer appreciated the author's responses to the comments.

1. Definition of entry to the open/closed arm

The author: "Time 0 s was set when the front half of the mouse's body enters the open or closed arm, which has been clarified in the Methods (Page 36, Line 720-721)."

The reviewer: The well-accepted definition of an arm entry is "all four paws must enter the arm." The author is requested to 1) report the results by recomputing using the strict definition, 2) add results on the distance to the center of mass of animals from a landmark of the maze vs. time from all subjects together with the average. In addition, when animals are in the half-body at the boundary, particularly at the entry of the open arm, they hesitate to go to the open arm. The author is requested to 3) report data in both cases: they go to the open arm and return to the closed arm.

We appreciated the reviewers for these highly valuable inputs. In the revised version,

we have followed the reviewer's suggestion and 1) reanalyzed the data following the definition of an arm entry with four paws entering the arm (**Figure 7**); 2) added results on the distance to the center of mass of animals from a landmark of the maze vs. time from all subjects together with the average (**Supplementary Figure 9**); 3) reported the data when the mice hesitated to go to the open arms (**Supplementary Figure 10**).

2. Ruling out the possibility of contribution from collateral projections

The reviewer: CNQX and AP5 infusion into the BNST is anxiolytic (Kim et al., 2013, see below). The author provided new results, supplementary Fig6, which indicate no anxiolytic effect with the treatment in the open field and elevated plus maze. The reviewer understands implanting the optical cannula (200um diameter) and the cannula for infusion (26G RWD, 457um diameter) is challenging. The associated histology does not match the expected tissue damage from optical and infusion cannulae (657um).

The author is requested to explain the two discrepancies above and describe the details of the implants.

Kim, Sung-Yon, Avishek Adhikari, Soo Yeun Lee, James H. Marshel, Christina K. Kim, Caitlin S. Mallory, Maisie Lo, et al. "Diverging Neural Pathways Assemble a Behavioural State from Separable Features in Anxiety." *Nature* 496, no. 7444 (April 2013): 219–23. <https://doi.org/10.1038/nature12018>.

The exact reasons are incompletely clear for the 1st discrepancy regarding the effect of CNQX and AP5 infusion in dBNST. However, the different animals used between the two studies may contribute to the discrepancy. While the animals used by Kim et al. were housed individually with high anxiety, those used in our study were normally housed (4-5 mice per cage) with relatively low anxiety.

For the 2nd discrepancy, we are sorry for not providing detailed introduction of the methodology. Actually, we only implanted a guide cannula (26 G) to the dBNST for both drug infusion and delivery of light stimuli. To deliver blue light through the guide

cannula, the optical fiber was initially shell-removed and passed through the cannula to the dBNST (Page 27; Line 540-544). Thus, the tissue damage was only caused by the guide cannula. We attached the chart introduction of the methods for reviewer's reference.

3. Data handling of optometry results in Figure 2

The reviewer restates the concern "Fig2m, n, p and q. In Fig2n and q, the difference of averaged %dF/F for 60 min is small relative to the peak heights in the traces". The author's raw photometry data shows an unstable baseline as regular, originating from photobleaching and locomotion. Calibration of the base with curve/linear fitting inevitably included a substantial error when comparing the values from averaged signals, particularly when the author sought to detect small differences (0.3 in Fig2n, 0.7 in Fig2q). The author is requested to report the average amplitudes and frequencies of the transients to avoid such errors.

Thanks for the valuable advices. We have provided the data in **Figure 2**. (**Figure 2o, p, t, u**; Page 35, Line 709-712).

REVIEWERS' COMMENTS

Reviewer #2 (Remarks to the Author):

The reviewer appreciated the author's responses to the comments.

The author is requested to include the discussion on the discrepancy with the report from the Deisseroth lab for credibility and fix the title of "Supplementary Figure 9," which is supposed to be "Supplementary Figure 11".

Once the corrections are done, there is no concern from the reviewer.

Responses to the reviewers' comments

Reviewer #2 (Remarks to the Author):

The reviewer appreciated the author's responses to the comments.

The author is requested to include the discussion on the discrepancy with the report from the Deisseroth lab for credibility and fix the title of "Supplementary Figure 9," which is supposed to be "Supplementary Figure 11".

Once the corrections are done, there is no concern from the reviewer.

We have discussed the discrepancy in our revised manuscript (Page 8, Line 178-181). The title of "Supplementary Figure 9," has been corrected to "Supplementary Figure 11".